# Concentration of Data Encoding in Parameterized Quantum Circuits

**Guangxi Li**[1,2], **Ruilin Ye**[2,3], **Xuanqiang Zhao**[2], **Xin Wang**[2*]

[1] University of Technology Sydney, NSW, Australia
[2] Institute for Quantum Computing, Baidu Research, Beijing, China
[3] Peking University, Beijing, China

## Abstract

Variational quantum algorithms have been acknowledged as the leading strategy to realize near-term quantum advantages in meaningful tasks, including machine learning and optimization. When applied to tasks involving classical data, such algorithms generally begin with data encoding circuits and train quantum neural networks (QNNs) to minimize target functions. Although QNNs have been widely studied to improve these algorithms' performance on practical tasks, there is a gap in systematically understanding the influence of data encoding on the eventual performance. In this paper, we make progress in filling this gap by considering the common data encoding strategies based on parameterized quantum circuits. We prove that, under reasonable assumptions, the distance between the average encoded state and the maximally mixed state could be explicitly upper-bounded with respect to the width and depth of the encoding circuit. This result in particular implies that the average encoded state will concentrate on the maximally mixed state at an exponential speed on depth. Such concentration seriously limits the capabilities of quantum classifiers, and strictly restricts the distinguishability of encoded states from a quantum information perspective. To support our findings, we numerically verify these results on both synthetic and public data sets. Our results highlight the significance of quantum data encoding and may shed light on the future design of quantum encoding strategies.

## 1 Introduction

Quantum machine learning [1, 2, 3, 4] is an emerging and promising interdisciplinary research direction in the fields of quantum computing and artificial intelligence. In this area, quantum computers are expected to enhance machine learning algorithms through their inherent parallel characteristics, thus demonstrating quantum advantages over classical algorithms [5].

With the increasing enormous efforts from academia and industry, the current quantum devices (usually acknowledged as the *noisy intermediate-scale quantum* (NISQ) devices [6]) already can show quantum advantages on certain carefully designed tasks [7, 8] despite their limitations in quantum circuit width and depth. A timely direction is to explore quantum advantages with near-term quantum devices in practical machine learning tasks, and a leading strategy is the hybrid quantum-classical algorithms, also known as the variational quantum algorithms [9, 10]. Such algorithms usually use a classical optimizer to train *quantum neural networks* (QNNs). They in particular hand over relatively intractable tasks to quantum computers while those rather ordinary tasks to classical computers.

---

*wangxin73@baidu.com

36th Conference on Neural Information Processing Systems (NeurIPS 2022).

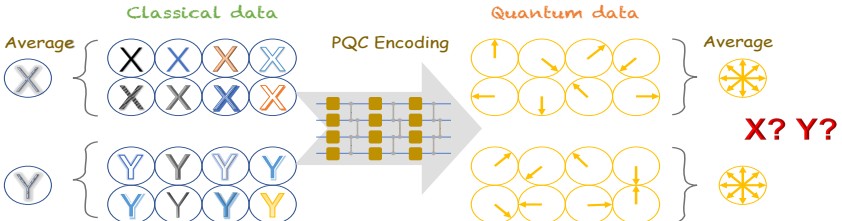

Figure 1: Cartoon illustrating the concentration of PQC-based data encoding. The average encoded quantum states concentrate on the maximally mixed state at an exponential rate on the encoding depth. This concentration implies the theoretical indistinguishability of the encoded quantum data.

For usual quantum machine learning tasks, a quantum circuit used in the variational quantum algorithms is generally divided into two parts: a data encoding circuit and a QNN. On the one hand, developing various QNN architectures is the most popular way to improve these algorithms' ability to deal with practical tasks. Numerous architectures such as strongly entangling circuit architectures [11], quantum convolutional neural networks [12], tree-tensor networks [13], and even automatically searched architectures [14, 15, 16, 17] have been proposed. On the other hand, one has to carefully design the encoding circuit, which could significantly influence the generalization performance of these algorithms [18, 19]. Consider an extreme case. If all classical inputs are encoded into the same quantum state, these algorithms will fail to do any machine learning tasks. In addition, the kernel's perspective [20, 21, 22, 23] also suggests that data encoding strategy plays a vital or even leading role in quantum machine learning algorithms [24, 25, 26]. However, there is much less literature on data encoding strategies, which urgently requires to be studied.

Encoding classical information into quantum data is nontrivial [1], and it is even more difficult on near-term quantum devices. One of the most feasible and popular encoding strategies on NISQ devices is based on *parameterized quantum circuits* (PQCs) [27], such as the empirically designed angle encoding [24, 28], IQP encoding [29], etc. It is natural to ask how to choose these encoding strategies and whether there are theoretical guarantees of using them. More specifically, it is necessary to systematically understand the impact of such PQC-based encoding strategies on the performance of QNNs in practical quantum machine learning tasks.

In this work, we present results from the perspective of average encoded quantum state with respect to the width and depth of the encoding PQCs. A cartoon illustration summarizing our main result is depicted in Fig. 1. We show that for the usual PQC-based encoding strategies with a fixed width, the average encoded state is close to the maximally mixed state at an exponential speed on depth. The contributions of this paper are multi-fold:

- We theoretically give the upper bound of the quantum divergence between the average encoded state and the maximally mixed state, which depends explicitly on the hyper-parameters (e.g., qubit number and encoding depth) of PQCs. From this bound, we find that for a fixed qubit number, the average encoded state concentrates on the maximally mixed state exponentially on the encoding depth.

- We show that the quantum states encoded by deep PQCs will seriously limit the trainability of a quantum classifier and further limit its classification ability.

- We show that the quantum states encoded by deep PQCs are indistinguishable from a quantum information perspective.

- We support the above findings by numerical experiments on both synthetic and public data sets.

**Related Work**   Ref. [18] derived generalization bounds of PQC-based data encoding, which mainly depends on the total number of circuit gates. While we derive quantum divergence bound that depends on the width and depth of PQCs. The works [26, 30] explored the effects of data encoding from the perspective of data re-uploading. [31] studied the robustness of data encoding for quantum classifiers. Data encoding strategies with discrete features were proposed for variational quantum classifiers [32].

## 1.1 Background

**Quantum Basics** To better understand this paper, some basic concepts about quantum computing [33] are provided here. In general, quantum information is described by quantum states. An $n$-qubit quantum state is mathematically represented by a positive semi-definite matrix (a.k.a. a density matrix) $\rho \in \mathbb{C}^{2^n \times 2^n}$ with property $\text{Tr}(\rho) = 1$. If $\text{Rank}(\rho) = 1$, it is called a pure state; otherwise, it is a mixed state. A pure state can also be represented by a unit column vector $|\phi\rangle \in \mathbb{C}^{2^n}$, where $\rho = |\phi\rangle\langle\phi|$ and $\langle\phi| = |\phi\rangle^\dagger$. A mixed state can be regarded as a weighted sum of pure states, i.e., $\rho = \sum_i q_i |\phi_i\rangle\langle\phi_i|$, where $q_i \geq 0$ and $\sum_i q_i = 1$. Specifically, a mixed state whose density matrix is proportional to the identity matrix is called the maximally mixed state $\mathbb{1} \equiv \frac{I}{2^n}$.

A quantum state $\rho$ could be evolved to another state $\rho'$ through a quantum circuit (or gate) mathematically represented by a unitary matrix $U$, i.e., $\rho' = U\rho U^\dagger$. Typical single-qubit gates include Pauli gates, $X \equiv \begin{bmatrix} 0 & 1 \\ 1 & 0 \end{bmatrix}, Y \equiv \begin{bmatrix} 0 & -i \\ i & 0 \end{bmatrix}, Z \equiv \begin{bmatrix} 1 & 0 \\ 0 & -1 \end{bmatrix}$, and their corresponding rotation gates $R_P(\theta) \equiv \text{e}^{-i\theta P/2}$ with a parameter $\theta$ and $P \in \{X, Y, Z\}$. Another commonly used gate $U3$ appeared in this paper is defined as $U3(\theta_1, \theta_2, \theta_3) \equiv R_z(\theta_3)R_y(\theta_2)R_z(\theta_1)$, which can implement an arbitrary single-qubit unitary transformation with appropriate parameters. In this paper, $R_z, R_y$ are equivalent to $R_Z, R_Y$ without specified. A multi-qubit gate can be either an individual gate (e.g., CNOT) or a tensor product of single-qubit gates. To get classical information from quantum state $\rho'$, one needs to perform quantum measurements, e.g., calculating the expectation value $\langle H \rangle = \text{Tr}(H\rho')$ of a Hermitian matrix $H$, and we often call $H$ an observable.

**Quantum Divergence** Similar to Kullback-Leibler divergence in machine learning, we use quantum divergence to quantify the difference between quantum states or quantum data. Two widely-used quantum divergences are quantum sandwiched Rényi divergence [34, 35] $\widetilde{D}_\alpha(\rho\|\sigma) \equiv \frac{1}{\alpha-1} \log \text{Tr}\left[\sigma^{\frac{1-\alpha}{2\alpha}} \rho \sigma^{\frac{1-\alpha}{2\alpha}}\right]^\alpha$ and the Petz-Rényi divergence [36] $D_\alpha(\rho\|\sigma) \equiv \frac{1}{\alpha-1} \log \text{Tr}\left[\rho^\alpha \sigma^{1-\alpha}\right]$, where $\alpha \in (0, 1) \cup (1, \infty)$ and the latter has an operational significance in quantum hypothesis testing [37, 38, 39]. In this work, for the purposes of analyzing quantum encoding, we focus on the Petz-Rényi divergence with $\alpha = 2$, i.e.,

$$D_2(\rho\|\sigma) = \log \text{Tr}\left[\rho^2 \sigma^{-1}\right], \tag{1}$$

which also plays an important role in training quantum neural networks [40] as well as quantum communication [41]. Throughout this paper, when we mention the quantum divergence, we mean the Petz-Rényi divergence $D_2$ if not specified; $\log$ denotes $\log_2$ if not specified.

**Parameterized Quantum Circuit** In general, a parameterized quantum circuit [27] has the form $U(\boldsymbol{\theta}) = \prod_j U_j(\theta_j)V_j$, where $\boldsymbol{\theta}$ is its parameter vector, $U_j(\theta_j) = \text{e}^{-i\theta_j P_j/2}$ with $P_j$ denoting a Pauli gate, and $V_j$ denotes a fixed gate such as Identity, CNOT and so on. In this paper, PQCs are utilized as both data encoding strategies and quantum neural networks. Specifically, when used for data encoding, an $n$-qubit PQC takes a classical input vector $\boldsymbol{x}$ as its parameters and acts on an initial state $|0\rangle^{\otimes n}$ to obtain the encoded state $|\boldsymbol{x}\rangle$. Here, $|0\rangle^{\otimes n}$ is a $2^n$-dimensional vector whose first element is 1 and all other elements are 0.

## 2 Main Results

### 2.1 A Warm-up Case

For a quick access, we first consider one of the most straightforward PQC-based data encoding strategies, i.e., consisting of $R_y$ rotations only, cf. Fig. 2(a). It can be viewed as a generalized angle encoding. For a classical input vector $\boldsymbol{x}$ with $nD$ components, the output of this data encoding circuit is a pure state $|\boldsymbol{x}\rangle \in \mathbb{C}^{2^n}$ expanded in a $2^n$-dimensional Hilbert space. We denote the density matrix of the output state by $\rho(\boldsymbol{x}) = |\boldsymbol{x}\rangle\langle\boldsymbol{x}|$. If we assume each element of the input vector obeys an *independent Gaussian distribution* (IGD, see Fig. 2(b) for an intuitive illustration), then we have the following theorem.

**Theorem 1.** *Assume each element of an $nD$-dimensional vector $\boldsymbol{x}$ obeys an IGD, i.e., $x_{j,d} \sim \mathcal{N}(\mu_{j,d}, \sigma_{j,d}^2)$, where $\sigma_{j,d} \geq \sigma$ for some constant $\sigma$ and $1 \leq j \leq n, 1 \leq d \leq D$. If $\boldsymbol{x}$ is encoded into*

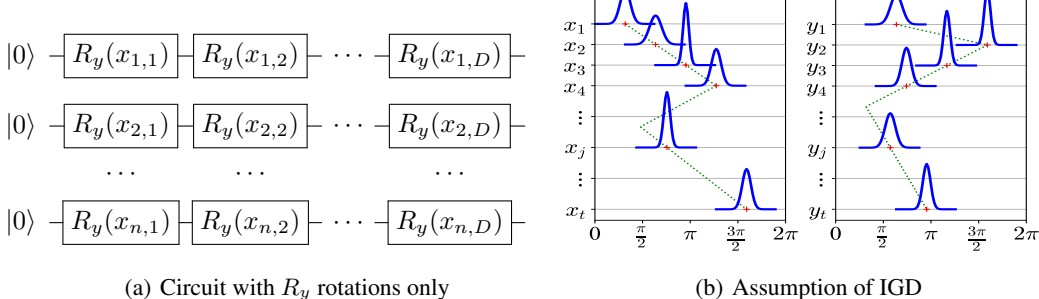

(a) Circuit with $R_y$ rotations only

(b) Assumption of IGD

Figure 2: (a) Circuit for the data encoding strategy with $R_y$ rotations only; (b) An example of a binary data set with two classes of $t$-dimensional vectors $\boldsymbol{x}$ and $\boldsymbol{y}$. Here, it is assumed that each $x_j$ (or $y_j$) obeys an independent Gaussian distribution (IGD), i.e., $x_j \sim \mathcal{N}(\mu_{x,j}, \sigma_{x,j}^2)$ (or $y_j \sim \mathcal{N}(\mu_{y,j}, \sigma_{y,j}^2)$), where these mean values (small red cross symbols) range in $[0, 2\pi)$ and form the green dotted lines. Note that the difference between these two lines determines that they belong to different classes.

*an $n$-qubit pure state $\rho(\boldsymbol{x})$ according to the circuit in Fig. 2(a), then the quantum divergence between the average encoded state $\bar{\rho}$ and the maximally mixed state $\mathbb{1}$ is upper-bounded as*

$$D_2\left(\bar{\rho}\|\mathbb{1}\right) \leq n\log\left(1 + \mathrm{e}^{-D\sigma^2}\right), \tag{2}$$

*where $\bar{\rho}$ is defined as $\bar{\rho} \equiv \mathbb{E}\left[\rho(\boldsymbol{x})\right]$.*

This theorem shows that the upper bound of the quantum divergence between $\bar{\rho}$ and $\mathbb{1}$ explicitly depends on the qubit number $n$ and the encoding depth $D$ under certain conditions. Further by approximating Eq. (2) as

$$D_2\left(\bar{\rho}\|\mathbb{1}\right) \leq n\log(1 + \mathrm{e}^{-D\sigma^2}) \approx \begin{cases} n\left(1 - \frac{\sigma^2}{2\ln 2}D\right), & D \in O(1) \\ n\mathrm{e}^{-D\sigma^2}, & D \in \Omega(\mathrm{poly}\log(n)) \end{cases}, \tag{3}$$

we easily find that for a fixed $n$, the upper bound decays exponentially with $D$ growing in $\Omega(\mathrm{poly}\log(n))$. This means that the average encoded state will quickly approach the maximally mixed state with an arbitrarily small distance under reasonable depths.

**Sketch of Proof.** For the $j$-th qubit, let $x_{j,\mathrm{sum}} = \sum_d x_{j,d}$ and $\mu_{j,\mathrm{sum}} = \sum_d \mu_{j,d}$, we get $x_{j,\mathrm{sum}} \sim \mathcal{N}(\mu_{j,\mathrm{sum}}, \sum_d \sigma_{j,d}^2)$. From the fact that $R_y(x_1)R_y(x_2) = R_y(x_1 + x_2)$, we have $\rho\left(\boldsymbol{x}_j\right) = R_y\left(x_{j,\mathrm{sum}}\right)|0\rangle\langle 0| R_y^\dagger\left(x_{j,\mathrm{sum}}\right) = \frac{1}{2}\left[I + \cos\left(x_{j,\mathrm{sum}}\right)Z + \sin\left(x_{j,\mathrm{sum}}\right)X\right]$. Further from the fact that if a variable $x \sim \mathcal{N}(\mu, \sigma^2)$, then $\mathbb{E}\left[\cos(x)\right] = \mathrm{e}^{-\frac{\sigma^2}{2}}\cos(\mu)$ and $\mathbb{E}\left[\sin(x)\right] = \mathrm{e}^{-\frac{\sigma^2}{2}}\sin(\mu)$, together with the condition $\sigma_{j,d} \geq \sigma$, we bound $D_2\left(\mathbb{E}\left[\rho\left(\boldsymbol{x}_j\right)\right]\|\mathbb{1}\right)$ via calculating

$$\mathrm{Tr}\left(\mathbb{E}^2\left[\rho\left(\boldsymbol{x}_j\right)\right]\right) = \frac{1}{2}\left(1 + \mathrm{e}^{-\sum_d \sigma_{j,d}^2}\right) \leq \frac{1}{2}\left(1 + \mathrm{e}^{-D\sigma^2}\right). \tag{4}$$

Finally we generalize Eq. (4) to multi-qubit cases, and complete the proof of Theorem 1. The detailed proof is deferred to Appendix A.

## 2.2 General Case

Next, we consider the general PQC-based data encoding strategies shown in Fig. 3, where a column of $U3$ gates and a column of entangled gates spread out alternately.

**Theorem 2.** *(Data Encoding Concentration) Assume each element of a $3nD$-dimensional vector $\boldsymbol{x}$ obeys an IGD, i.e., $x_{j,d,k} \sim \mathcal{N}(\mu_{j,d,k}, \sigma_{j,d,k}^2)$, where $\sigma_{j,d,k} \geq \sigma$ for some constant $\sigma$ and $1 \leq j \leq n, 1 \leq d \leq D, 1 \leq k \leq 3$. If $\boldsymbol{x}$ is encoded into an $n$-qubit pure state $\rho(\boldsymbol{x})$ according to the circuit in Fig. 3, the quantum divergence between the average encoded state $\bar{\rho}$ and the maximally mixed state $\mathbb{1}$ is upper-bounded as*

$$D_2\left(\bar{\rho}\|\mathbb{1}\right) \leq \log(1 + (2^n - 1)\mathrm{e}^{-D\sigma^2}). \tag{5}$$

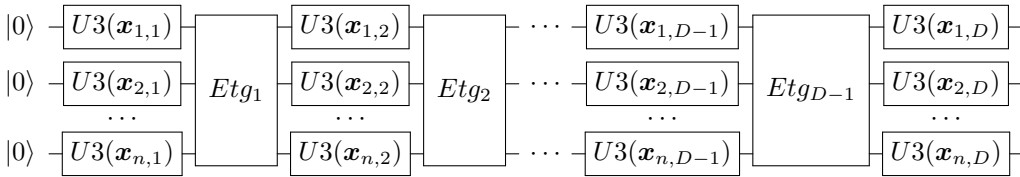

Figure 3: Circuit for the data encoding strategy with $D$ layers of $U3$ gates and $D-1$ layers of entanglements. Here, each $\boldsymbol{x}_{j,d}$ represents three elements $x_{j,d,1}, x_{j,d,2}, x_{j,d,3}$, and each $Etg_i$ denotes an arbitrary group of entangled two-qubit gates, such as CNOT or CZ, where $1 \leq j \leq n, 1 \leq d \leq D, 1 \leq i \leq D-1$.

This theorem shows that, when employing general PQC-based encoding strategies, we could also have an upper bound of the quantum divergence $D_2\left(\bar{\rho}\|\mathbb{1}\right)$ which explicitly depends on $n$ and $D$. By approximating the upper bound in Eq. (5) as follows

$$D_2\left(\bar{\rho}\|\mathbb{1}\right) \leq \log\left(1 + (2^n - 1)\mathrm{e}^{-D\sigma^2}\right) \approx \begin{cases} n - \frac{\sigma^2}{\ln 2}D, & D \in O(\mathrm{poly}\log(n)) \\ (2^n - 1)\mathrm{e}^{-D\sigma^2}, & D \in \Omega(\mathrm{poly}(n)) \end{cases}, \quad (6)$$

we observe similarly that for some fixed $n$, the upper bound decays at an exponential speed as $D$ grows in $\Omega(\mathrm{poly}(n))$. In addition, according to our proof analysis, even if each $U3$ gate is replaced by a $U2$ gate containing only two different kinds of Pauli rotations or even a $U1$ gate with only one proper Pauli rotation, we still get a similar bound as Eq. (5). Therefore, we conclude that as long as $D$ grows within a reasonable scope, the average of the quantum states encoded by a wide family of PQCs will quickly concentrates on the maximally mixed state. Unlike the warm-up case, the proof for this theorem is quite non-straightforward due to the tricky entangled gates. For smooth reading, we defer the complete proof in Appendix B.

The average encoded state $\bar{\rho}$ in Theorem 2 is built on infinite data samples, but in practice we do not have infinite ones. Therefore, we provide the following helpful corollary.

**Corollary 2.1.** *Assume there are $M$ classical vectors $\{\boldsymbol{x}^{(m)}\}_{m=1}^M$ sampled from the distributions described in Theorem 2 and define $\bar{\rho}_M \equiv \frac{1}{M}\sum_{m=1}^M \rho(\boldsymbol{x}^{(m)})$. Let $H$ be a Hermitian matrix with its eigenvalues ranging in $[-1, 1]$, then given an arbitrary $\epsilon \in (0, 1)$, as long as the encoding depth $D \geq \frac{1}{\sigma^2}\left[(n+4)\ln 2 + 2\ln(1/\epsilon)\right]$, we have*

$$\left| \mathrm{Tr}\left[H\left(\bar{\rho}_M - \mathbb{1}\right)\right] \right| \leq \epsilon \quad (7)$$

*with a probability of at least $1 - 2\mathrm{e}^{-M\epsilon^2/8}$.*

This corollary implies that for a reasonable encoding depth $D$ and number of samples $M$, the practical average encoded state $\bar{\rho}_M$ will also be infinitely close to the maximally mixed state with a high probability. The proof is mainly derived from *Hoeffding's inequality* [42] and the relations between quantum divergence and trace norm. The proof details are deferred to Appendix C.

## 3 Applications in Quantum Supervised Learning

In this section, we will show that the concentrated quantum states encoded by the above PQC-based data encoding strategies will severely limit the performances of quantum supervised learning tasks. Before beginning, let's define the following necessary data set.

**Definition 3.** *(Data Set) The $K$-class data set $\mathcal{D} \equiv \{(\boldsymbol{x}^{(m)}, \boldsymbol{y}^{(m)})\}_{m=1}^{KM} \subset \mathbb{R}^{3nD} \times \mathbb{R}^K$ totally has $KM$ data samples, including $M$ samples in each category. Here, suppose elements in the same entry of all input vectors from the same category are sampled from the same IGD with a variance of at least $\sigma^2$ and each $\boldsymbol{x}^{(m)}$ is encoded into the corresponding pure state $\rho(\boldsymbol{x}^{(m)})$ according to the circuit in Fig. 3 with $n$ qubits and $D$ layers of $U3$ gates. The label $\boldsymbol{y}^{(m)}$ is a one-hot vector that indicates which of the $K$ classes $\boldsymbol{x}^{(m)}$ belongs to.*

## 3.1 In the Scenario of Quantum Classification

Quantum classification, as one of the most significant branches in quantum machine learning, is widely studied nowadays, where quantum-enhanced classification models are expected to achieve quantum advantages against the classical ones in solving classification tasks [29, 43]. Specifically, in the NISQ era, plenty of variational quantum classifiers based on parameterized quantum circuits are developed to make full use of NISQ devices [9, 44, 11, 12, 13, 45, 46]. See [47] for a review on the recent advances of quantum classifiers.

In general, a quantum classifier aims to learn a map from input to label by optimizing a loss function constructed through QNNs to predict the label of an unseen input as accurately as possible. Now, we demonstrate the performance of a quantum classifier on the data set $\mathcal{D}$ defined in Def. 3.

In this paper, the loss function is defined from the cross-entropy loss with softmax function [48]:

$$L\left(\boldsymbol{\theta};\mathcal{D}\right) \equiv \frac{1}{KM}\sum_{m=1}^{KM} L^{(m)} \quad \text{with} \quad L^{(m)}\left(\boldsymbol{\theta};(\boldsymbol{x}^{(m)},\boldsymbol{y}^{(m)})\right) \equiv -\sum_{k=1}^{K} y_k^{(m)} \ln \frac{\mathrm{e}^{h_k}}{\sum_{j=1}^{K}\mathrm{e}^{h_j}}, \quad (8)$$

where $y_k^{(m)}$ denotes the $k$-th element of the label $\boldsymbol{y}^{(m)}$ and

$$h_k\left(\boldsymbol{x}^{(m)},\boldsymbol{\theta}\right) = \mathrm{Tr}\left[H_k U(\boldsymbol{\theta})\rho(\boldsymbol{x}^{(m)})U^{\dagger}(\boldsymbol{\theta})\right], \quad (9)$$

which means the Hermitian operator $H_k$ is finally measured after the quantum neural network $U(\boldsymbol{\theta})$. Here, each $H_k$ is chosen from tensor products of various Pauli matrices, such as $Z \otimes I$, $X \otimes Y \otimes Z$ and so on. By minimizing the loss function with a gradient descent method, we could obtain the final trained model $U(\boldsymbol{\theta}^*)$ with the optimal or sub-optimal parameters $\boldsymbol{\theta}^*$. After that, when provided a new input quantum state $\rho(\boldsymbol{x}')$, we compute each $h_k'$ with parameters $\boldsymbol{\theta}^*$ according to Eq. (9), and the index of the largest $h_k'$ is exactly our designated label.

However, all these graceful expectations can only be established on gradients with relatively large absolute values. On the contrary, gradients with significantly small absolute values will cause a severe training problem, for example the barren plateau issue [49]. Therefore, in the following we investigate the partial gradient of the cost defined in Eq. (8) with regard to its parameters. The results are exhibited in Proposition 4.

**Proposition 4.** *Consider a $K$-classification task with the data set $\mathcal{D}$ defined in Def. 3. If the encoding depth $D \geq \frac{1}{\sigma^2}\left[(n+4)\ln 2 + 2\ln(1/\epsilon)\right]$ for some $\epsilon \in (0,1)$, then the partial gradient of the loss function defined in Eq. (8) with respect to each parameter $\theta_i$ of the employed QNN is bounded as*

$$\left|\frac{\partial L\left(\boldsymbol{\theta};\mathcal{D}\right)}{\partial \theta_i}\right| \leq K\epsilon \quad (10)$$

*with a probability of at least $1 - 2\mathrm{e}^{-M\epsilon^2/8}$.*

From this proposition, we observe that no matter what QNN structures are selected, the absolute gradient value can be arbitrarily small with a very high probability for the above data set $\mathcal{D}$, provided that the encoding depth $D$ and the number of data samples $M$ are sufficiently large. This vanishing of the gradients will severely restrict the trainability of QNNs. Moreover, if before training $U(\boldsymbol{\theta})$ is initialized to satisfy a certain randomness, such as unitary 2-design [50], then each $h_k$ in Eq. (9) will concentrate on 0 with a high probability, thus the loss in Eq. (8) will concentrate on $\ln K$. This concentration of loss is also verified through numerical simulations, as presented in Sec. 4. This phenomenon, together with Proposition 4, implies that large encoding depth will significantly hinder the training of a quantum classifier and probably lead to poor classification accuracy. Please refer to Appendix D for the proof of Proposition 4.

## 3.2 In the Scenario of Quantum State Discrimination

Quantum state discrimination [51] is a central information-theoretic task and finds applications in various topics such as quantum cryptography [52], quantum error mitigation [53], and quantum data hiding [54]. It aims to distinguish quantum states using a positive operator-valued measure (POVM), a set of positive semi-definite operators that sum to the identity operator. Here, we have to seriously note that in quantum state discrimination, we can only measure each quantum state once, instead of measuring repeatedly and calculating the expectations as shown in quantum classification.

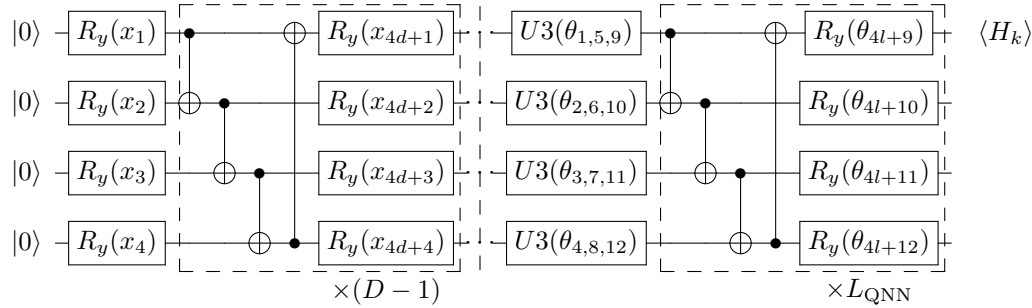

Figure 4: Circuits for data encoding (before the barrier line) and quantum neural network (after the barrier line) in the 4-qubit case. Here the input $\boldsymbol{x} \in \mathbb{R}^{4D}$ and $d \in [1, D-1]$. The QNN totally has $L_{\mathrm{QNN}} + 1$ layers with parameters $\boldsymbol{\theta} \in \mathbb{R}^{4L_{\mathrm{QNN}}+12}$, where the first layer $U3$ gates consists of 12 parameters. After QNN, there are $K$ expectations $\{\langle H_k \rangle\}_{k=1}^{K}$ for $K$-classification tasks.

In general, a perfect discrimination (i.e., a perfect POVM) can not be achieved if quantum states are non-orthogonal. A natural alternative option is by adopting some metrics such as the success probability so that the optimal POVM could be obtained via various kinds of optimization ways, e.g., Helstrom bound [55] and semi-definite programming (SDP) [56]. Recently, researchers also try to train QNNs as substitutions for optimal POVMs [57, 58].

Next, we demonstrate the impact of the encoded quantum states from the data set $\mathcal{D}$ defined in Def. 3 on quantum state discrimination. Our goal is to obtain the maximum success probability $p_{\mathrm{succ}}$ by maximizing the success probability over all POVMs with $K$ operators:

$$p_{\mathrm{succ}} \equiv \max_{\{\Pi_k\}_k} \frac{1}{K} \sum_{k=1}^{K} \mathrm{Tr}\left[\Pi_k \bar{\rho}_{k,M}\right] \quad \text{with} \quad \bar{\rho}_{k,M} \equiv \frac{1}{M} \sum_{m=1}^{KM} y_k^{(m)} \rho(\boldsymbol{x}^{(m)}), \tag{11}$$

where $y_k^{(m)}$ denotes the $k$-th element of the label $\boldsymbol{y}^{(m)}$ and $\{\Pi_k\}_{k=1}^{K}$ denotes a POVM, which satisfies $\sum_{k=1}^{K} \Pi_k = I$.

**Proposition 5.** *Consider a $K$-class discrimination task with the data set $\mathcal{D}$ defined in Def. 3. If the encoding depth $D \geq \frac{1}{\sigma^2}\left[(n+4)\ln 2 + 2\ln(1/\epsilon)\right]$ for a given $\epsilon \in (0,1)$, then with a probability of at least $1 - 2\mathrm{e}^{-M\epsilon^2/8}$, the maximum success probability $p_{\mathrm{succ}}$ is bounded as*

$$p_{\mathrm{succ}} \leq 1/K + \epsilon. \tag{12}$$

This proposition implies that as long as the encoding depth $D$ and the data numbers $M$ are large enough, the optimal success probability $p_{\mathrm{succ}}$ could be arbitrarily close to $\frac{1}{K}$ with a remarkably high probability for the data set $\mathcal{D}$. This nearly blind-guessing success probability shows that the concentration of the encoded quantum states in the data set $\mathcal{D}$ will lead to the failure of state discrimination via POVM. As POVMs are the most general kind of measurements one can implement to extract classical information from quantum systems [33], we conclude that the above different classes of encoded states are indistinguishable from the perspective of quantum information. The proof of Proposition 5 could be derived straightforwardly by combining Eq. (11) with Corollary 2.1.

## 4   Numerical Experiments

Previous sections demonstrate that the average encoded state will concentrate on the maximally mixed state under PQC-based data encoding strategies with large depth. These encoded states theoretically cannot be utilized to train QNNs or distinguished by POVMs. In this section, we verify these results on both synthetic and public data sets by choosing a commonly employed strongly entangling circuit [11], which helps to understand the concentration rate intuitively for realistic encoding circuits. All the simulations and optimization loop are implemented via Paddle Quantum[2] on the PaddlePaddle Deep Learning Platform [59].

---

[2] https://github.com/paddlepaddle/Quantum

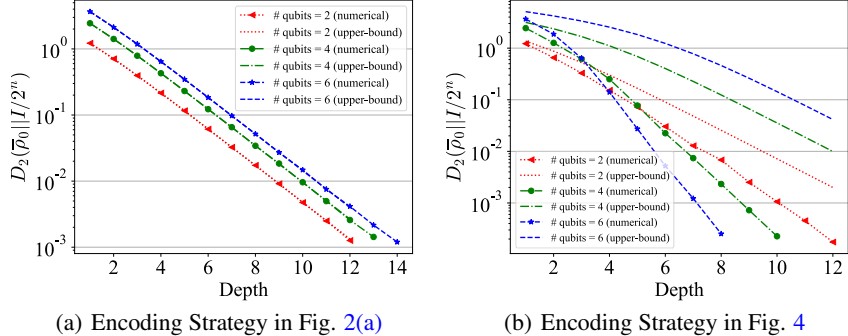

(a) Encoding Strategy in Fig. 2(a)  (b) Encoding Strategy in Fig. 4

Figure 5: Exponential decay of quantum divergence $D_2(\bar{\rho}_0 || \mathbb{1})$ vs. encoding depth under different qubit cases for synthetic data set. Here, there are one million data samples for calculating average encoded state $\bar{\rho}_0$ for class 0 at each point in numerical lines. And the upper-bounds come from (a) Theorem 1, (b) Theorem 2, respectively.

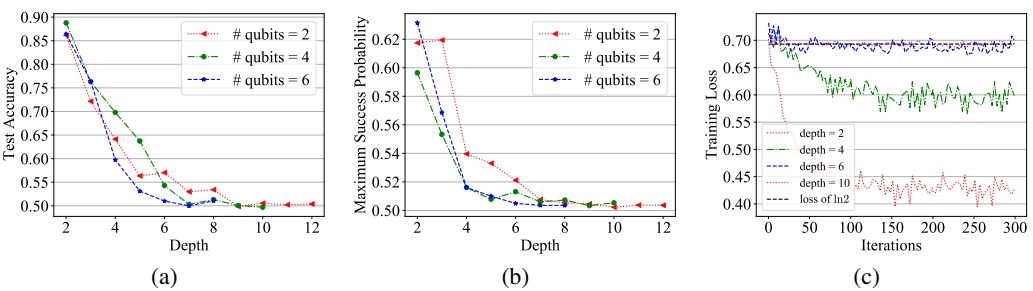

(a)  (b)  (c)

Figure 6: Numerical results for synthetic data sets under the encoding strategy in Fig. 4. In all qubit cases, (a) the test accuracy of QNN (or (b) the maximum success probability of POVM) will eventually decay to 50% or so with the depth growing; (c) In the 4 qubit case, for instance, the training losses of QNN do not decrease and stay at about $\ln 2$ in the training process when the depth becomes large enough.

## 4.1 On Synthetic Data Set

**Data Set**  The synthetic two-class data set $\left\{ (\boldsymbol{x}^{(m)}, \boldsymbol{y}^{(m)}) \right\}_{m=1}^{M}$ is generated following the distributions depicted in Fig. 2(b), where each $x_j^{(m)} \sim \mathcal{N}(\mu_j, \sigma_j^2)$ for $1 \leq j \leq t$ and $\boldsymbol{y}^{(m)}$ denotes a one-hot label. Here, we assume all means come from two lines, i.e., $\mu_j = \frac{2\pi}{16}(j-1) \bmod 2\pi$ for class 0 and $\mu_j = \frac{2\pi}{16}(16-j) \bmod 2\pi$ for class 1, and all $\sigma_j$'s are set as 0.8. Note that the same variance is selected for both classes to facilitate the demonstration of the experiment. Other choices of $\sigma_j$'s would have similar effects.

We first verify our two main upper bounds given in Theorems 1 and 2 by encoding the $nD$-dimensional inputs that belong to the same class into $n$-qubit quantum states with $D$ encoding depths under the encoding strategies illustrated in Figs. 2(a) and 4, respectively. Here, $n$ is set as $2, 4, 6$ and $D \in [1, 14]$. The results are displayed in Fig. 5, from which we can intuitively see that the divergences decrease exponentially on depth. Specifically, from Fig. 5(a), we know the upper bound in Theorem 2(a) is tight, which is also easily verified from our proof. From Fig. 5(b), we learn that the upper bound in Theorem 2 is quite loose, which suggests that the real situation is much worse than our theoretical analysis. We also notice in Fig. 5(b) that for this strongly entangling encoding strategy, the larger the qubit number is, the faster the divergence decreases. This unexpected phenomenon reveals the possibility that specific structures of encoding circuits may lead to more severe concentrations for larger numbers of qubits and is worthy of further studies.

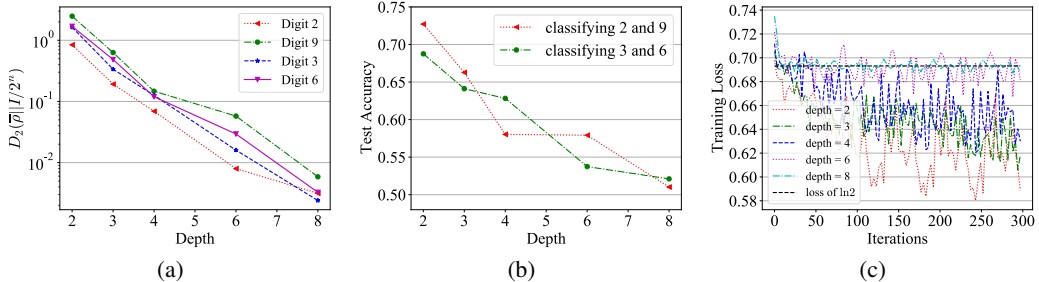

Figure 7: Numerical results of QNN for MNIST data set under the encoding strategy in Fig. 4. (a) The curves for the quantum divergence between the averaged encoded state $\bar{\rho}$ of each handwritten digit and the maximally mixed state $\mathbb{1}$ decrease exponentially on depth. (b) The test accuracy reduce rapidly with a larger encoding depth; (c) In the case of classifying digits 3 and 6, when the depth is large (e.g., 8), it is difficult to keep the training loss away from $\ln 2$ in the training process.

Next, we examine the performance of QNNs and POVMs by generating 20k data samples for training and 4k for testing under the encoding strategy in Fig. 4, where half of the data belong to class 0, and the others belong to class 1. The QNNs are designed according to the right hand side of Fig. 4, where the number of layers $L_{\mathrm{QNN}}$ is set as $n + 2$, the finally measured Hermitian operators are set as $H_1 = Z$ and $H_2 = X$ on the first qubit, and all parameters $\boldsymbol{\theta}$ are initialized randomly in $[0, 2\pi]$. During the optimization, we adopt the Adam optimizer [60] with a batch size of 200 and a learning rate of $0.02$. In the POVM setting, we directly employ semi-definite programming [56] to obtain the maximum success probability $P_{\mathrm{succ}}$ on the training data samples. The results are illustrated in Fig. 6. We observe that both the test accuracy of the QNNs and the maximum success probability $P_{\mathrm{succ}}$ of the POVMs eventually decay to about $0.5$ as the encoding depth grows, indicating that the classification abilities of both the QNNs and the POVMs are no better than random guessing. In addition, the training losses of QNNs in Fig. 6(c) gradually approach $\ln 2$ as the depth grows and finally do not go down anymore during the whole training process, which implies that the concentration of this data set on the maximally mixed state would limit the trainability of QNNs as we predicted in Sec. 3.1. All these results are in line with our theoretical expectations.

### 4.2 On Public Data Set

**Data Set and Preprocessing** The handwritten digit data set MNIST [61] consists of 70k images labeled from '0' to '9', each of which contains $28 \times 28$ gray scale pixels valued in $[0, 255]$. In order to facilitate encoding, these images are first resized to $4 \times 4$ and then normalized to values between $0$ and $\pi$. Finally, we select all images corresponding to two pairs of labels, i.e., $(2, 9)$ and $(3, 6)$, for two binary classification tasks. For each task, there are about 12k training samples and 2k testing samples, and each category accounts for half or so.

Here we mainly consider the performance of QNN on this data set because POVMs are generally not suitable for prediction. These 16-dimensional preprocessed images are first encoded into $n$-qubit quantum states with encoding depth $D$ and then fed into a QNN (cf. Fig. 4 again). We set $n$ as 2,3,4,6,8 and $D$ as 8,6,4,3,2 accordingly. The settings of QNN are almost the same as those used in the synthetic case, except for a new learning rate of $0.05$. From Fig. 7(a), we see that the average state of each digit class concentrates on the maximally mixed state at an approximately exponential speed on depth, which is consistent with our main result. Furthermore, the outcomes in Figs. 7(b) and 7(c) also confirm the incapability of training of QNNs, provided that the classical inputs are encoded by a higher depth PQC.

## 5 Discussion

We have witnessed both theoretically and numerically that for usual PQC-based data encoding strategies with higher depth, the average encoded state concentrates on the maximally mixed state. We further show that such concentration severely limits the capabilities of quantum classifiers for

practical tasks. Such limitation indicates that we should pay more attention to methods encoding classical data into PQCs in quantum supervised learning.

Our work suggests that the distance between the average encoded state and the maximally mixed state may be a reasonable metric to quantify how well the quantum encoding preserves the features in quantum supervised learning. The result on the encoding concentration also motivates us to consider how to design PQC-based encoding strategies better to avoid the exponentially decayed distance. An obvious way this paper implies might be to keep the depth shallow while accompanied by a higher width. Still, it will render poor generalization performance [19] as well as the notorious barren plateau issue [49]. Therefore, it will be desirable to develop nontrivial quantum encoding strategies to guarantee the effectiveness and efficiency of quantum supervised learning as well as quantum kernel methods [24, 20, 28]. Amplitude encoding [24] may serve as a good candidate, but due to its unavailability on NISQ devices, it is rarely considered or implemented at current stage. Recent works on data re-uploading [26, 30, 62] and pooling [19, 25] of quantum neural networks may also provide potential methods for improving quantum encoding efficiency.

## Acknowledgements

We would like to thank Sanjiang Li, Yuan Feng, Hongshun Yao, and Zhan Yu for helpful discussions. G. L. acknowledges the support from the Baidu-UTS AI Meets Quantum project, the China Scholarship Council (No. 201806070139), and the Australian Research Council project (Grant No: DP180100691). Part of this work was done when R. Y. and X. Z. were research interns at Baidu Research.

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
