# Supplementary Material for "Concentration of Data Encoding in Parameterized Quantum Circuits"

## A    Proof of Theorem 1

**Theorem 1.** *Assume each element of an $nD$-dimensional vector $\boldsymbol{x}$ obeys an IGD, i.e., $x_{j,d} \sim \mathcal{N}(\mu_{j,d}, \sigma_{j,d}^2)$, where $\sigma_{j,d} \geq \sigma$ for some constant $\sigma$ and $1 \leq j \leq n, 1 \leq d \leq D$. If $\boldsymbol{x}$ is encoded into an $n$-qubit pure state $\rho(\boldsymbol{x})$ according to the circuit in Fig. S1, then the quantum divergence between the average encoded state $\bar{\rho} = \mathbb{E}\left[\rho(\boldsymbol{x})\right]$ and the maximally mixed state $\mathbb{1} = \frac{I}{2^n}$ is upper bounded as*

$$D_2\left(\bar{\rho} \| \mathbb{1}\right) \equiv \log \mathrm{Tr}\left(\bar{\rho}^2 \cdot \mathbb{1}^{-1}\right) \leq n \log\left(1 + \mathrm{e}^{-D\sigma^2}\right). \tag{S1}$$

Figure S1: Circuit for the data encoding strategy with $R_y$ rotations only.

*Proof.* Let $\rho(\boldsymbol{x}_j) \equiv R_y(x_{j,1} + \cdots + x_{j,D}) |0\rangle\langle 0| R_y^\dagger(x_{j,1} + \cdots + x_{j,D})$, then

$$\rho(\boldsymbol{x}) = \rho(\boldsymbol{x}_1) \otimes \rho(\boldsymbol{x}_2) \otimes \cdots \otimes \rho(\boldsymbol{x}_n). \tag{S2}$$

Due to the independence of each $\rho(\boldsymbol{x}_j)$, we have

$$\bar{\rho} = \mathbb{E}\left[\rho(\boldsymbol{x})\right] = \mathbb{E}\left[\rho(\boldsymbol{x}_1)\right] \otimes \mathbb{E}\left[\rho(\boldsymbol{x}_2)\right] \otimes \cdots \otimes \mathbb{E}\left[\rho(\boldsymbol{x}_n)\right]. \tag{S3}$$

What's more, for $j = 1, \ldots, n$,

$$\mathbb{E}\left[\rho(\boldsymbol{x}_j)\right] = \frac{1}{2}\mathbb{E}\begin{bmatrix} 1 + \cos\left(\sum_d x_{j,d}\right) & \sin\left(\sum_d x_{j,d}\right) \\ \sin\left(\sum_d x_{j,d}\right) & 1 - \cos\left(\sum_d x_{j,d}\right) \end{bmatrix} \tag{S4}$$

$$= \frac{1}{2}\begin{bmatrix} 1 + \mathbb{E}\left[\cos\left(\sum_d x_{j,d}\right)\right] & \mathbb{E}\left[\sin\left(\sum_d x_{j,d}\right)\right] \\ \mathbb{E}\left[\sin\left(\sum_d x_{j,d}\right)\right] & 1 - \mathbb{E}\left[\cos\left(\sum_d x_{j,d}\right)\right] \end{bmatrix}. \tag{S5}$$

**Lemma 1.** *Assume a variable $x \sim \mathcal{N}(\mu, \sigma^2)$, then*

$$\mathbb{E}\left[\cos(x)\right] = \mathrm{e}^{-\frac{\sigma^2}{2}} \cos(\mu); \qquad \mathbb{E}\left[\sin(x)\right] = \mathrm{e}^{-\frac{\sigma^2}{2}} \sin(\mu). \tag{S6}$$

We know $\sum_d x_{j,d} \sim \mathcal{N}(\sum_d \mu_{j,d}, \sqrt{\sum_d \sigma_{j,d}^2})$, and combining Eq. (S5) with Lemma 1, we have

$$\left\| \mathbb{E}\left[\rho(\boldsymbol{x}_j)\right] \right\|_F^2 = \frac{1}{2} + \frac{1}{2}\mathbb{E}^2\left[\cos\left(\sum_d x_{j,d}\right)\right] + \frac{1}{2}\mathbb{E}^2\left[\sin\left(\sum_d x_{j,d}\right)\right] \tag{S7}$$

$$= \frac{1}{2} + \frac{1}{2}\left(\mathrm{e}^{-\frac{\sum_d \sigma_{j,d}^2}{2}} \cos(\sum_d \mu_{j,d})\right)^2 + \frac{1}{2}\left(\mathrm{e}^{-\frac{\sum_d \sigma_{j,d}^2}{2}} \sin(\sum_d \mu_{j,d})\right)^2 \tag{S8}$$

$$= \frac{1}{2} + \frac{1}{2}\mathrm{e}^{-\sum_d \sigma_{j,d}^2} \tag{S9}$$

$$\leq \frac{1}{2} + \frac{1}{2}\mathrm{e}^{-D\sigma^2}. \tag{S10}$$

Finally, from Eq. (S3), we have

$$\log \operatorname{Tr}\left(\bar{\rho}^2 \cdot \left(\frac{I}{2^n}\right)^{-1}\right) = \log\left(2^n \|\bar{\rho}\|_F^2\right) \tag{S11}$$

$$= \log\left(2^n \prod_{j=1}^{n} \left\|\mathbb{E}\left[\rho(\boldsymbol{x}_j)\right]\right\|_F^2\right) \tag{S12}$$

$$\leq \log\left(2^n \left(\frac{1 + \mathrm{e}^{-D\sigma^2}}{2}\right)^n\right) \tag{S13}$$

$$= n \log(1 + \mathrm{e}^{-D\sigma^2}). \tag{S14}$$

This completes the proof. $\qquad\square$

## B  Proof of Theorem 2

**Theorem 2.** *Assume each element of a $3nD$-dimensional vector $\boldsymbol{x}$ obeys an IGD, i.e., $x_{j,d,k} \sim \mathcal{N}(\mu_{j,d,k}, \sigma_{j,d,k}^2)$, where $\sigma_{j,d,k} \geq \sigma$ for some constant $\sigma$ and $1 \leq j \leq n, 1 \leq d \leq D, 1 \leq k \leq 3$. If $\boldsymbol{x}$ is encoded into an $n$-qubit pure state $\rho(\boldsymbol{x})$ according to the circuit in Fig. S2, the quantum divergence between the average encoded state $\bar{\rho}$ and the maximally mixed state $\mathbb{1}$ is upper bounded as*

$$D_2\left(\bar{\rho}\|\mathbb{1}\right) \equiv \log \operatorname{Tr}\left(\bar{\rho}^2 \cdot \mathbb{1}^{-1}\right) \leq \log\left(1 + (2^n - 1)\mathrm{e}^{-D\sigma^2}\right). \tag{S15}$$

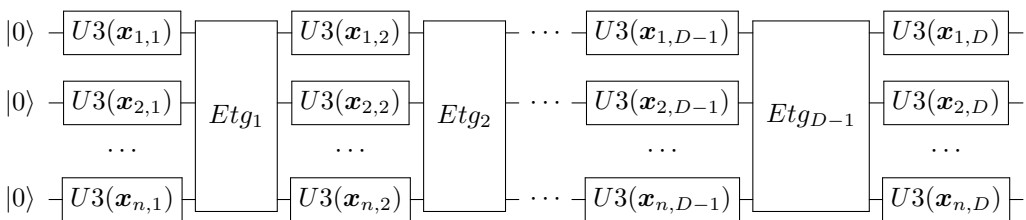

Figure S2: Circuit for the data encoding strategy with $D$ layers of $U3$ gates and $D-1$ layers of entanglements. Here, each $\boldsymbol{x}_{j,d}$ represents three elements $x_{j,d,1}, x_{j,d,2}, x_{j,d,3}$, and each $Etg_i$ denotes an arbitrary group of entangled two-qubit gates, such as CNOT or CZ, where $1 \leq j \leq n, 1 \leq d \leq D, 1 \leq i \leq D - 1$.

*Proof.* In this proof, we consider the $U3(x_{j,d,1}, x_{j,d,2}, x_{j,d,3})$ gate as $R_z(x_{j,d,3}) \cdot R_y(x_{j,d,2}) \cdot R_z(x_{j,d,1})$, which is one of the most commonly used ones. Of course, other forms of U3 gate are similar.

**Outline of Proof.** **1) Decomposing initial state.** Firstly, we decompose the initial state according to Pauli bases; **2) Vectors transition.** Then by taking the corresponding coefficients as a row vector, we state that each action of a group of entangled gates $Etg_i$ or a column of $U3$ gates is equivalent to multiplying the previous coefficient vector by a transition matrix; **3) Bound by singular value.** Finally, we get the upper bound by investigating the singular values of these transition matrices.

**1) Decomposing initial state.** The state after the first column of $U3$ gates becomes

$$\rho_1 = \frac{1}{2}\begin{bmatrix} 1 + \cos(x_{1,1,2}) & \mathrm{e}^{-ix_{1,1,3}}\sin(x_{1,1,2}) \\ \mathrm{e}^{ix_{1,1,3}}\sin(x_{1,1,2}) & 1 - \cos(x_{1,1,2}) \end{bmatrix} \otimes \frac{1}{2}\begin{bmatrix} 1 + \cos(x_{2,1,2}) & \mathrm{e}^{-ix_{2,1,3}}\sin(x_{2,1,2}) \\ \mathrm{e}^{ix_{2,1,3}}\sin(x_{2,1,2}) & 1 - \cos(x_{2,1,2}) \end{bmatrix}$$
$$\otimes \cdots \cdots \otimes \frac{1}{2}\begin{bmatrix} 1 + \cos(x_{n,1,2}) & \mathrm{e}^{-ix_{n,1,3}}\sin(x_{n,1,2}) \\ \mathrm{e}^{ix_{n,1,3}}\sin(x_{n,1,2}) & 1 - \cos(x_{n,1,2}) \end{bmatrix}. \tag{S16}$$

Table 1: The transition table for tensor products of Pauli bases when applying CNOT or CZ gates.

| Pauli bases | Apply CNOT | Apply CZ |
|:-:|:-:|:-:|
| $I \otimes I$ | $I \otimes I$ | $I \otimes I$ |
| $I \otimes Z$ | $Z \otimes Z$ | $I \otimes Z$ |
| $I \otimes X$ | $I \otimes X$ | $Z \otimes X$ |
| $I \otimes Y$ | $Z \otimes Y$ | $Z \otimes Y$ |
| $Z \otimes I$ | $Z \otimes I$ | $Z \otimes I$ |
| $Z \otimes Z$ | $I \otimes Z$ | $Z \otimes Z$ |
| $Z \otimes X$ | $Z \otimes X$ | $I \otimes X$ |
| $Z \otimes Y$ | $I \otimes Y$ | $I \otimes Y$ |
| $X \otimes I$ | $X \otimes X$ | $X \otimes Z$ |
| $X \otimes Z$ | $-Y \otimes Y$ | $X \otimes I$ |
| $X \otimes X$ | $X \otimes I$ | $Y \otimes Y$ |
| $X \otimes Y$ | $Y \otimes Z$ | $-Y \otimes X$ |
| $Y \otimes I$ | $Y \otimes X$ | $Y \otimes Z$ |
| $Y \otimes Z$ | $X \otimes Y$ | $Y \otimes I$ |
| $Y \otimes X$ | $Y \otimes I$ | $-X \otimes Y$ |
| $Y \otimes Y$ | $-X \otimes Z$ | $X \otimes X$ |

Now we define $\rho_1 \equiv \rho_{1,1} \otimes \rho_{1,2} \otimes \cdots \otimes \rho_{1,n}$, where

$$\rho_{1,j} \equiv \frac{1}{2} \begin{bmatrix} 1 + \cos(x_{j,1,2}) & \mathrm{e}^{-ix_{j,1,3}} \sin(x_{j,1,2}) \\ \mathrm{e}^{ix_{j,1,3}} \sin(x_{j,1,2}) & 1 - \cos(x_{j,1,2}) \end{bmatrix}. \tag{S17}$$

And from Lemma 1, we have

$$\mathbb{E}\left[\rho_{1,j}\right] = \frac{1}{2} \begin{bmatrix} 1 + A_{j,1,2}\cos(\mu_{j,1,2}) & A_{j,1,3}\mathrm{e}^{-i\mu_{j,1,3}}A_{j,1,2}\sin(\mu_{j,1,2}) \\ A_{j,1,3}\mathrm{e}^{i\mu_{j,1,3}}A_{j,1,2}\sin(\mu_{j,1,2}) & 1 - A_{j,1,2}\cos(\mu_{j,1,2}) \end{bmatrix}, \tag{S18}$$

where we define $A_{j,d,k} = \mathrm{e}^{-\frac{\sigma_{j,d,k}^2}{2}}$ for writing convenience. Here we note that due to all $x_{j,d,k}$'s being independent of each other, calculating the expectation of $\rho_{1,j}$ in advance does not affect the following computations. Next we decompose $\mathbb{E}\left[\rho_{1,j}\right]$ according to the Pauli bases, i.e., $I, Z, X, Y$,

$$\mathbb{E}\left[\rho_{1,j}\right] = \frac{1}{2}I + \frac{A_{j,1,2}\cos(\mu_{j,1,2})}{2}Z + \frac{A_{j,1,3}\cos(\mu_{j,1,3})A_{j,1,2}\sin(\mu_{j,1,2})}{2}X$$
$$+ \frac{A_{j,1,3}\sin(\mu_{j,1,3})A_{j,1,2}\sin(\mu_{j,1,2})}{2}Y. \tag{S19}$$

Then from Eq. (S16), we could derive that $\mathbb{E}\left[\rho_1\right] = \bigotimes_{j=1}^{n} \mathbb{E}\left[\rho_{1,j}\right]$ can also be decomposed in accordance with various tensor products of Pauli bases. Therefore, studying the state after the gate $Etg_1$ could be transferred to what performance it will be when entangled two-qubit gates act on the tensor products of Pauli bases.

**2) Vectors transition.** Here, we focus on the two widely employed two-qubit entangled gates CNOT and CZ, and the calculations are concluded in Table 1. The results in the table show that the transitions are closed for tensor products of Pauli bases. Here we note that other entangled two-qubit gates will have a similar effect.

Next, we consider the effects of applying the gate $U3(x_1, x_2, x_3) = R_z(x_3)R_y(x_2)R_z(x_1)$ to four Pauli matrices. And the results of the calculations are as follows:

$$\mathbb{E}\left[U3 \cdot I \cdot U3^\dagger\right] = I \tag{S20}$$

$$\mathbb{E}\left[U3 \cdot Z \cdot U3^\dagger\right] = p_{zz}Z + p_{zx}X + p_{zy}Y \tag{S21}$$

$$\mathbb{E}\left[U3 \cdot X \cdot U3^\dagger\right] = p_{xz}Z + p_{xx}X + p_{xy}Y \tag{S22}$$

$$\mathbb{E}\left[U3 \cdot Y \cdot U3^\dagger\right] = p_{yz}Z + p_{yx}X + p_{yy}Y, \tag{S23}$$

where

$$p_{zz} = A_2 \cos(\mu_2) \tag{S24}$$
$$p_{zx} = A_2 \sin(\mu_2) A_3 \cos(\mu_3) \tag{S25}$$
$$p_{zy} = A_2 \sin(\mu_2) A_3 \sin(\mu_3) \tag{S26}$$
$$p_{xz} = -A_2 \sin(\mu_2) A_1 \cos(\mu_1) \tag{S27}$$
$$p_{xx} = A_2 \cos(\mu_2) A_1 \cos(\mu_1) A_3 \cos(\mu_3) - A_1 \sin(\mu_1) A_3 \sin(\mu_3) \tag{S28}$$
$$p_{xy} = A_2 \cos(\mu_2) A_1 \cos(\mu_1) A_3 \sin(\mu_3) + A_1 \sin(\mu_1) A_3 \cos(\mu_3) \tag{S29}$$
$$p_{yz} = A_2 \sin(\mu_2) A_1 \sin(\mu_1) \tag{S30}$$
$$p_{yx} = -A_2 \cos(\mu_2) A_1 \sin(\mu_1) A_3 \cos(\mu_3) - A_1 \cos(\mu_1) A_3 \sin(\mu_3) \tag{S31}$$
$$p_{yy} = -A_2 \cos(\mu_2) A_1 \sin(\mu_1) A_3 \sin(\mu_3) + A_1 \cos(\mu_1) A_3 \cos(\mu_3). \tag{S32}$$

Here, $x_k, A_k, \mu_k$ are the abbreviations for $x_{j,d,k}, A_{j,d,k}, \mu_{j,d,k}$, respectively.

Now we record Eqs. (S20)-(S23) as a matrix $T$, which we call *transition matrix*, i.e.,

$$T \equiv \begin{bmatrix} 1 & 0 & 0 & 0 \\ 0 & p_{zz} & p_{zx} & p_{zy} \\ 0 & p_{xz} & p_{xx} & p_{xy} \\ 0 & p_{yz} & p_{yx} & p_{yy} \end{bmatrix}. \tag{S33}$$

By carefully calculating Eqs. (S20)-(S33) again, we also have

$$T = \begin{bmatrix} 1 & 0 & 0 & 0 \\ 0 & 1 & 0 & 0 \\ 0 & 0 & A_1\cos(\mu_1) & A_1\sin(\mu_1) \\ 0 & 0 & -A_1\sin(\mu_1) & A_1\cos(\mu_1) \end{bmatrix} \begin{bmatrix} 1 & 0 & 0 & 0 \\ 0 & A_2\cos(\mu_2) & A_2\sin(\mu_2) & 0 \\ 0 & -A_2\sin(\mu_2) & A_2\cos(\mu_2) & 0 \\ 0 & 0 & 0 & 1 \end{bmatrix} \begin{bmatrix} 1 & 0 & 0 & 0 \\ 0 & 1 & 0 & 0 \\ 0 & 0 & A_3\cos(\mu_3) & A_3\sin(\mu_3) \\ 0 & 0 & -A_3\sin(\mu_3) & A_3\cos(\mu_3) \end{bmatrix}, \tag{S34}$$

where the three matrices correspond to the effects of applying $R_z(x_1)$, $R_y(x_2)$ and $R_z(x_3)$, respectively.

If we further record an arbitrary input $\rho_{in} \equiv \alpha_1 I + \alpha_2 Z + \alpha_3 X + \alpha_4 Y$ as a row vector $\pi_{in} = [\alpha_1 \quad \alpha_2 \quad \alpha_3 \quad \alpha_4]$, then applying the gate $U3(x_1, x_2, x_3)$ to $\rho_{in}$ will result in the output $\rho_{out} \equiv \beta_1 I + \beta_2 Z + \beta_3 X + \beta_4 Y$, where

$$\pi_{out} \equiv [\beta_1 \quad \beta_2 \quad \beta_3 \quad \beta_4] = [\alpha_1 \quad \alpha_2 \quad \alpha_3 \quad \alpha_4] \begin{bmatrix} 1 & 0 & 0 & 0 \\ 0 & p_{zz} & p_{zx} & p_{zy} \\ 0 & p_{xz} & p_{xx} & p_{xy} \\ 0 & p_{yz} & p_{yx} & p_{yy} \end{bmatrix} = \pi_{in}T. \tag{S35}$$

This is a fundamental relationship in this proof, which can be easily verified in multi-qubit and multi-depth cases. Hence, we could rewrite each $\mathbb{E}[\rho_d]$, $0 \leq d \leq D$, as follows

$$\mathbb{E}[\rho_0] \quad \longleftrightarrow \quad \pi_0 = \otimes_{j=1}^{n} \begin{bmatrix} \frac{1}{2} & \frac{1}{2} & 0 & 0 \end{bmatrix} \tag{S36}$$

$$\mathbb{E}[\rho_1] \quad \longleftrightarrow \quad \pi_1 = \pi_0 \cdot \otimes_{j=1}^{n} T_{j,1} \cdot \widetilde{Etg_1} \tag{S37}$$

$$\cdots$$

$$\mathbb{E}[\rho_d] \quad \longleftrightarrow \quad \pi_d = \pi_{d-1} \cdot \otimes_{j=1}^{n} T_{j,d} \cdot \widetilde{Etg_d} \tag{S38}$$

$$\cdots$$

$$\mathbb{E}[\rho_{D-1}] \quad \longleftrightarrow \quad \pi_{D-1} = \pi_{D-2} \cdot \otimes_{j=1}^{n} T_{j,D-1} \cdot \widetilde{Etg_{D-1}} \tag{S39}$$

$$\mathbb{E}[\rho_D] \quad \longleftrightarrow \quad \pi_D = \pi_{D-1} \cdot \otimes_{j=1}^{n} T_{j,D}, \tag{S40}$$

where each $T_{j,d}$ represents that this transition matrix is constructed based on the gate $U3(x_{j,d,1}, x_{j,d,2}, x_{j,d,3})$ and each $\widetilde{Etg_i}$ means rearranging the elements of the previously multiplied row vector, which is equivalent to the effect after applying $Etg_i$, $1 \leq i \leq D-1$. Here, please note that we omit the possible negative sign described in Table 1, because in the following proof, it has no influence.

From the fact that $\text{Tr}(P_i^2) = 2$, $\text{Tr}(P_i P_j) = 0$, where $P_i, P_j$ denote different Pauli matrices, and combining the relationship in Eq. (S40), we have

$$\text{Tr}(\mathbb{E}[\rho_D])^2 = 2^n \cdot \pi_D (\pi_D)^\top. \tag{S41}$$

What's more, we also find that every $\otimes_{j=1}^n T_{j,d}$ and $\widetilde{Etg_i}$ always have an element 1 in the top left corner, i.e.,

$$\otimes_{j=1}^n T_{j,d} \equiv \begin{bmatrix} 1 & \\ & \mathcal{T}_d \end{bmatrix}, \qquad \widetilde{Etg_i} \equiv \begin{bmatrix} 1 & \\ & \mathcal{E}_i \end{bmatrix}, \tag{S42}$$

where $\mathcal{T}_d, \mathcal{E}_i \in \mathbb{R}^{(4^n-1)\times(4^n-1)}$ and $1 \le d \le D, 1 \le i \le D-1$. Therefore,

$$\mathrm{Tr}\left(\mathbb{E}\left[\rho_D\right]\right)^2 = 2^n \cdot \pi_D \left(\pi_D\right)^\top \tag{S43}$$

$$= 2^n \cdot \pi_0 \begin{bmatrix} 1 & \\ & \mathcal{T}_1\mathcal{E}_1\mathcal{T}_2\mathcal{E}_2\cdots\mathcal{T}_D \end{bmatrix} \begin{bmatrix} 1 & \\ & \mathcal{T}_D^\top\cdots\mathcal{E}_2^\top\mathcal{T}_2^\top\mathcal{E}_1^\top\mathcal{T}_1^\top \end{bmatrix} \left(\pi_0\right)^\top \tag{S44}$$

$$= 2^n \cdot \begin{bmatrix} \frac{1}{2^n} & \mathring{\pi}_0 \end{bmatrix} \begin{bmatrix} 1 & \\ & \mathcal{T}_1\mathcal{E}_1\mathcal{T}_2\mathcal{E}_2\cdots\mathcal{T}_D\mathcal{T}_D^\top\cdots\mathcal{E}_2^\top\mathcal{T}_2^\top\mathcal{E}_1^\top\mathcal{T}_1^\top \end{bmatrix} \begin{bmatrix} \frac{1}{2^n} \\ \mathring{\pi}_0^\top \end{bmatrix} \tag{S45}$$

$$= \frac{1}{2^n} + 2^n \cdot \mathring{\pi}_0\mathcal{T}_1\mathcal{E}_1\cdots\mathcal{T}_{D-1}\mathcal{E}_{D-1}\mathcal{T}_D\mathcal{T}_D^\top\mathcal{E}_{D-1}^\top\mathcal{T}_{D-1}^\top\cdots\mathcal{E}_1^\top\mathcal{T}_1^\top\mathring{\pi}_0^\top, \tag{S46}$$

where $\mathring{\pi}_0$ means that the row vector $\pi_0$ removes the first element.

**3) Bound by singular value.** Next, in order to further calculate it, we need to prove first the following two Lemmas.

**Lemma 2.** *Given a Hermitian matrix $H \in \mathbb{C}^{n\times n}$ with all its eigenvalues no larger than $\lambda$, and an $n$-dimensional vector $\boldsymbol{x}$, then*

$$\boldsymbol{x}^\dagger H\boldsymbol{x} \le \|\boldsymbol{x}\|_2^2\lambda, \tag{S47}$$

*where $\|\cdot\|_2$ denotes the $l_2$-norm.*

*Proof.* Assume $H$ has the spectral decomposition

$$H = \sum_{i=1}^n \lambda_i \boldsymbol{u}_i\boldsymbol{u}_i^\dagger, \tag{S48}$$

then $\boldsymbol{x}$ can be uniquely decomposed as $\boldsymbol{x} = \sum_{i=1}^n \alpha_i\boldsymbol{u}_i$ with $\sum_{i=1}^n |\alpha_i|^2 = \|\boldsymbol{x}\|_2^2$. Finally we have

$$\boldsymbol{x}^\dagger H\boldsymbol{x} = \sum_{i=1}^n |\alpha_i|^2\lambda_i \le \sum_i^n |\alpha_i|^2\lambda = \|\boldsymbol{x}\|_2^2\lambda. \tag{S49}$$

$\square$

**Lemma 2.** *Given a Hermitian matrix $H \in \mathbb{C}^{n\times n}$ with all its eigenvalues no larger than $\lambda$, and an arbitrary matrix $Q \in \mathbb{C}^{n\times n}$ with all its singular values no larger than $s$, then the largest eigenvalue of $QHQ^\dagger$ is no larger than $s^2\lambda$.*

*Proof.* The largest eigenvalue of $QHQ^\dagger$ can be computed as $\lambda_{max} \equiv \max_{\boldsymbol{x}} \boldsymbol{x}^\dagger QHQ^\dagger\boldsymbol{x}$, where $\boldsymbol{x}$ denotes a unit vector. Assume $Q$ has the singular value decomposition

$$Q = USV^\dagger = \sum_{i=1}^n s_i\boldsymbol{u}_i\boldsymbol{v}_i^\dagger = \begin{bmatrix} \boldsymbol{u}_1 & \boldsymbol{u}_2 & \cdots & \boldsymbol{u}_n \end{bmatrix} \begin{bmatrix} s_1 & 0 & 0 & 0 \\ 0 & s_2 & 0 & 0 \\ 0 & 0 & \ddots & 0 \\ 0 & 0 & 0 & s_n \end{bmatrix} \begin{bmatrix} \boldsymbol{v}_1^\dagger \\ \boldsymbol{v}_2^\dagger \\ \vdots \\ \boldsymbol{v}_n^\dagger \end{bmatrix} \tag{S50}$$

and $\boldsymbol{x} = \sum_{i=1}^n \alpha_i\boldsymbol{u}_i$ with $\sum_{i=1}^n |\alpha_i|^2 = 1$, then

$$\boldsymbol{x}^\dagger QHQ^\dagger\boldsymbol{x} = \boldsymbol{x}^\dagger USV^\dagger HVSU^\dagger\boldsymbol{x} = \begin{bmatrix} \alpha_1^\dagger s_1 & \alpha_2^\dagger s_2 & \cdots & \alpha_n^\dagger s_n \end{bmatrix} V^\dagger HV \begin{bmatrix} \alpha_1 s_1 \\ \alpha_2 s_2 \\ \cdots \\ \alpha_n s_n \end{bmatrix}. \tag{S51}$$

Consider $V^\dagger H V$ as a new Hermitian matrix and $S U^\dagger \boldsymbol{x}$ as a new vector $\tilde{\boldsymbol{x}}$, then all the eigenvalues of $V^\dagger H V$ are still no larger than $\lambda$ and the square of the $l_2$-norm of $\tilde{\boldsymbol{x}}$ is computed as

$$\|\tilde{\boldsymbol{x}}\|_2^2 = \sum_{i=1}^n |\alpha_i|^2 s_i^2 \le \sum_{i=1}^n |\alpha_i|^2 s^2 = s^2. \tag{S52}$$

From Lemma 2, we have

$$\boldsymbol{x}^\dagger Q H Q^\dagger \boldsymbol{x} \le \|\tilde{\boldsymbol{x}}\|_2^2 \lambda \le s^2 \lambda. \tag{S53}$$

As $\boldsymbol{x}$ is arbitrary, we can obtain that $\lambda_{max} \equiv \max_{\boldsymbol{x}} \boldsymbol{x}^\dagger Q H Q^\dagger \boldsymbol{x}$ is no larger than $s^2 \lambda$ as well. $\qquad \square$

Now, let us investigate the singular values of $T_{j,d}$. From Eq. (S33), we know it always has the trivial biggest singular value 1. The second-biggest singular value $s_m$ can be derived from

$$s_m^2 = \max_{\boldsymbol{u}} \boldsymbol{u}^\dagger \begin{bmatrix} p_{zz} & p_{zx} & p_{zy} \\ p_{xz} & p_{xx} & p_{xy} \\ p_{yz} & p_{yx} & p_{yy} \end{bmatrix} \begin{bmatrix} p_{zz} & p_{xz} & p_{yz} \\ p_{zx} & p_{xx} & p_{yx} \\ p_{zy} & p_{xy} & p_{yy} \end{bmatrix} \boldsymbol{u}, \tag{S54}$$

where $\boldsymbol{u} \in \mathbb{C}^3$ denotes a unit column vector. From Eq. (S34), we derive that

$$\begin{bmatrix} p_{zz} & p_{zx} & p_{zy} \\ p_{xz} & p_{xx} & p_{xy} \\ p_{yz} & p_{yx} & p_{yy} \end{bmatrix} = \begin{bmatrix} 1 & 0 & 0 \\ 0 & A_1 \cos(\mu_1) & A_1 \sin(\mu_1) \\ 0 & -A_1 \sin(\mu_1) & A_1 \cos(\mu_1) \end{bmatrix} \begin{bmatrix} A_2 \cos(\mu_2) & A_2 \sin(\mu_2) & 0 \\ -A_2 \sin(\mu_2) & A_2 \cos(\mu_2) & 0 \\ 0 & 0 & 1 \end{bmatrix} \begin{bmatrix} 1 & 0 & 0 \\ 0 & A_3 \cos(\mu_3) & A_3 \sin(\mu_3) \\ 0 & -A_3 \sin(\mu_3) & A_3 \cos(\mu_3) \end{bmatrix} \tag{S55}$$

$$= \begin{bmatrix} 1 & 0 & 0 \\ 0 & A_1 \cos(\mu_1) & A_1 \sin(\mu_1) \\ 0 & -A_1 \sin(\mu_1) & A_1 \cos(\mu_1) \end{bmatrix} \begin{bmatrix} \cos(\mu_2) & \sin(\mu_2) & 0 \\ -\sin(\mu_2) & \cos(\mu_2) & 0 \\ 0 & 0 & 1 \end{bmatrix} \begin{bmatrix} A_2 & & \\ & A_2 A_3 & 0 \\ 0 & & A_3 \end{bmatrix} \begin{bmatrix} 1 & 0 & 0 \\ 0 & \cos(\mu_3) & \sin(\mu_3) \\ 0 & -\sin(\mu_3) & \cos(\mu_3) \end{bmatrix}, \tag{S56}$$

hence,

$$\begin{bmatrix} p_{zz} & p_{zx} & p_{zy} \\ p_{xz} & p_{xx} & p_{xy} \\ p_{yz} & p_{yx} & p_{yy} \end{bmatrix} \begin{bmatrix} p_{zz} & p_{xz} & p_{yz} \\ p_{zx} & p_{xx} & p_{yx} \\ p_{zy} & p_{xy} & p_{yy} \end{bmatrix} = Q \begin{bmatrix} \cos(\mu_2) & \sin(\mu_2) & 0 \\ -\sin(\mu_2) & \cos(\mu_2) & 0 \\ 0 & 0 & 1 \end{bmatrix} \begin{bmatrix} A_2^2 & & \\ & (A_2 A_3)^2 & 0 \\ 0 & & A_3^2 \end{bmatrix} \begin{bmatrix} \cos(\mu_2) & -\sin(\mu_2) & 0 \\ \sin(\mu_2) & \cos(\mu_2) & 0 \\ 0 & 0 & 1 \end{bmatrix} Q^\top, \tag{S57}$$

where $Q \equiv \begin{bmatrix} 1 & 0 & 0 \\ 0 & A_1 \cos(\mu_1) & A_1 \sin(\mu_1) \\ 0 & -A_1 \sin(\mu_1) & A_1 \cos(\mu_1) \end{bmatrix}$ has the largest singular value 1.

From Lemma 2, we deduce that the largest eigenvalue of the matrix in Eq. (S57) is $\max\{A_2^2, A_3^2\}$, which is no larger than $\mathrm{e}^{-\sigma^2}$. Further combining Eq. (S54), we infer that $s_m$ is no larger than $\mathrm{e}^{-\frac{\sigma^2}{2}}$, i.e., the second-biggest singular value of each $T_{j,d}$ is no larger than $\mathrm{e}^{-\frac{\sigma^2}{2}}$. What's more, we could derive that their tensor product $\otimes_{j=1}^n T_{j,d}$ also has the trivial largest singular value 1 and the second-largest singular value which is no larger than $\mathrm{e}^{-\frac{\sigma^2}{2}}$.

From the definition of $\mathcal{T}_d$ in Eq. (S42), we declare that the largest singular value of each $\mathcal{T}_d$ is no larger than $\mathrm{e}^{-\frac{\sigma^2}{2}}$. Let's go back to the following formula in Eq. (S46) to continue estimating $\mathrm{Tr}\left(\mathbb{E}\left[\rho_D\right]\right)^2$, i.e.,

$$\mathring{\pi}_0 \mathcal{T}_1 \mathcal{E}_1 \cdots \mathcal{T}_{D-1} \mathcal{E}_{D-1} \mathcal{T}_D \mathcal{T}_D^\top \mathcal{E}_{D-1}^\top \mathcal{T}_{D-1}^\top \cdots \mathcal{E}_1^\top \mathcal{T}_1^\top \mathring{\pi}_0^\top. \tag{S58}$$

Since the largest eigenvalue of $\mathcal{T}_D \mathcal{T}_D^\top$ is no larger than $\mathrm{e}^{-\sigma^2}$, and each $\mathcal{E}_i$, defined in Eq. (S42), is a unitary matrix, by repeatedly applying Lemma 2, we obtain that largest eigenvalue of $\mathcal{T}_1 \mathcal{E}_1 \cdots \mathcal{T}_D \mathcal{T}_D^\top \cdots \mathcal{E}_1^\top \mathcal{T}_1^\top$ is no larger than $\mathrm{e}^{-D\sigma^2}$. Furthermore, from Eq. (S36) and the definition of $\mathring{\pi}_0$, we know $\mathring{\pi}_0$ has $4^n - 1$ dimensions, where $2^n - 1$ elements are $\frac{1}{2^n}$ and the others are 0. Hence, $\|\mathring{\pi}_0\|_2^2 = \frac{2^n - 1}{2^{2n}}$. Combining these with Lemma 2, we have

$$\mathring{\pi}_0 \mathcal{T}_1 \mathcal{E}_1 \cdots \mathcal{T}_{D-1} \mathcal{E}_{D-1} \mathcal{T}_D \mathcal{T}_D^\top \mathcal{E}_{D-1}^\top \mathcal{T}_{D-1}^\top \cdots \mathcal{E}_1^\top \mathcal{T}_1^\top \mathring{\pi}_0^\top \le \|\mathring{\pi}_0\|_2^2 \mathrm{e}^{-D\sigma^2} = \frac{2^n - 1}{2^{2n}} \mathrm{e}^{-D\sigma^2}. \tag{S59}$$

Go further, and we have, together with Eq. (S46),

$$\mathrm{Tr}\left(\mathbb{E}\left[\rho_D\right]\right)^2 = \frac{1}{2^n} + 2^n \cdot \mathring{\pi}_0 \mathcal{T}_1 \mathcal{E}_1 \cdots \mathcal{T}_{D-1} \mathcal{E}_{D-1} \mathcal{T}_D \mathcal{T}_D^\top \mathcal{E}_{D-1}^\top \mathcal{T}_{D-1}^\top \cdots \mathcal{E}_1^\top \mathcal{T}_1^\top \mathring{\pi}_0^\top \tag{S60}$$

$$\le \frac{1}{2^n} + 2^n \cdot \frac{2^n - 1}{2^{2n}} \mathrm{e}^{-D\sigma^2} \tag{S61}$$

$$= \frac{1 + (2^n - 1)\,\mathrm{e}^{-D\sigma^2}}{2^n}. \tag{S62}$$

Finally, we have

$$\log \mathrm{Tr}\left(\bar{\rho}^2 \cdot \left(\frac{I}{2^n}\right)^{-1}\right) = \log\left(2^n \cdot \mathrm{Tr}\left(\mathbb{E}\left[\rho_D\right]\right)^2\right) \tag{S63}$$

$$\leq \log\left(1 + (2^n - 1)\mathrm{e}^{-D\sigma^2}\right). \tag{S64}$$

This completes the proof of Theorem 2.

Without stopping here, we also analyse some generalizations of Theorem 2.

(I) From Eqs. (S54)-(S57), we find that removing the matrix $Q$ will have no influence on the final result. Hence, we directly generalize that if there is only one column of $R_z R_y$ or $R_y R_z$ gates in each layer, we will get the same upper bound. In fact, according to our proof method, we infer that as long as there are two different kinds of rotation gates in each encoding layer, this upper bound is valid.

(II) What is the result for the case with only $R_y$ rotation gates in each encoding layer? Since each $\mathcal{T}_d$ has the largest singular value 1, it is not suitable for the above proof. However, through analyzing the transition rule in Table 1, we find that the largest singular value of $\mathcal{T}_{d-1}\mathcal{E}_{d-1}\mathcal{T}_d$ is still no larger than $\mathrm{e}^{-\frac{\sigma^2}{2}}$, which means every two encoding layers have the same effect as above with one layer. Therefore, the final upper bound can be changed to $\log\left(1 + (2^n - 1)\mathrm{e}^{-\lfloor\frac{D}{2}\rfloor\sigma^2}\right)$. Since it has the same trend as the original bound, it has no impact on our final analysis. □

## C   Proof of Corollary 2.1

**Corollary 2.1.** *Assume there are $M$ classical vectors $\{\boldsymbol{x}^{(m)}\}_{m=1}^M$ sampled from the distributions described in Theorem 2 and define $\bar{\rho}_M \equiv \frac{1}{M}\sum_{m=1}^M \rho(\boldsymbol{x}^{(m)})$. Let $H$ be a Hermitian matrix with its eigenvalues ranging in $[-1, 1]$, then given an arbitrary $\epsilon \in (0, 1)$, as long as the depth $D \geq \frac{1}{\sigma^2}\left[(n+4)\ln 2 + 2\ln(1/\epsilon)\right]$, we have*

$$\left|\mathrm{Tr}\left[H\left(\bar{\rho}_M - \mathbb{1}\right)\right]\right| \leq \epsilon \tag{S65}$$

*with a probability of at least $1 - 2\mathrm{e}^{-M\epsilon^2/8}$.*

*Proof.* Let $\bar{\rho} \equiv \mathbb{E}\left[\rho\left(\boldsymbol{x}^{(m)}\right)\right]$, then we have

$$\left|\mathrm{Tr}\left[H\left(\bar{\rho}_M - \frac{I}{2^n}\right)\right]\right| = \left|\mathrm{Tr}\left[H\left(\bar{\rho}_M - \bar{\rho} + \bar{\rho} - \frac{I}{2^n}\right)\right]\right| \tag{S66}$$

$$\leq \left|\mathrm{Tr}\left[H\left(\bar{\rho}_M - \bar{\rho}\right)\right]\right| + \left|\mathrm{Tr}\left[H\left(\bar{\rho} - \frac{I}{2^n}\right)\right]\right|, \tag{S67}$$

where the inequality is due to triangle inequality.

Now we first consider the first term in Eq. (S67). Since $\frac{1}{M}\mathrm{Tr}\left(H\rho(\boldsymbol{x}^{(1)})\right), \ldots, \frac{1}{M}\mathrm{Tr}\left(H\rho(\boldsymbol{x}^{(M)})\right)$ are i.i.d. and $\frac{-1}{M} \leq \frac{1}{M}\mathrm{Tr}\left(H\rho(\boldsymbol{x}^{(m)})\right) \leq \frac{1}{M}$, through *Hoeffding's inequality* [42], we have

$$P\left(\left|\sum_{m=1}^M \frac{1}{M}\mathrm{Tr}\left(H\rho(\boldsymbol{x}^{(m)})\right) - \mathbb{E}\left[\mathrm{Tr}\left(H\rho(\boldsymbol{x}^{(m)})\right)\right]\right| \leq t\right) \geq 1 - 2\mathrm{e}^{-\frac{Mt^2}{2}}. \tag{S68}$$

From the fact that $\sum_{m=1}^M \frac{1}{M}\mathrm{Tr}\left(H\rho(\boldsymbol{x}^{(m)})\right) = \mathrm{Tr}\left(H\bar{\rho}_M\right)$ and $\mathbb{E}\left[\mathrm{Tr}\left(H\rho(\boldsymbol{x}^{(m)})\right)\right] = \mathrm{Tr}\left(H\bar{\rho}\right)$, we obtain

$$\left|\mathrm{Tr}\left(H\bar{\rho}_M\right) - \mathrm{Tr}\left(H\bar{\rho}\right)\right| \leq \frac{\epsilon}{2} \tag{S69}$$

with a probability of at least $1 - 2\mathrm{e}^{-\frac{M\epsilon^2}{8}}$.

Next we consider the second term in Eq. (S67). Since the eigenvalues of $H$ range in $[-1, 1]$, we obtain

$$\left|\mathrm{Tr}\left(H\left(\bar{\rho} - \frac{I}{2^n}\right)\right)\right| \leq \left\|\bar{\rho} - \frac{I}{2^n}\right\|_{\mathrm{tr}} \leq 2\sqrt{1 - F\left(\bar{\rho}, \frac{I}{2^n}\right)}, \tag{S70}$$

where $\|\cdot\|_{\mathrm{tr}}$ denotes the trace norm and the second inequality is from the *Fuchs–van de Graaf inequalities* [63], i.e., $1 - \sqrt{F(\rho, \rho')} \leq \frac{1}{2}\|\rho - \rho'\|_{\mathrm{tr}} \leq \sqrt{1 - F(\rho, \rho')}$. By combining the upper bound in Theorem 2 with the fact that $-\log F(\rho, \rho') \leq D_2(\rho, \rho')$ [36], we have

$$F\left(\bar{\rho}, \frac{I}{2^n}\right) \geq \frac{1}{2^{D_2\left(\bar{\rho}\|\frac{I}{2^n}\right)}} \geq \frac{1}{1 + (2^n - 1)\mathrm{e}^{-D\sigma^2}} \tag{S71}$$

$$\geq \frac{1}{1 + \frac{(2^n - 1)\epsilon^2}{2^{n+4}}} \tag{S72}$$

$$\geq \frac{1}{1 + \frac{2^n \epsilon^2}{16 \cdot 2^n}} \tag{S73}$$

$$= \frac{16}{16 + \epsilon^2}, \tag{S74}$$

where in Eq. (S72) we use the condition $D \geq \frac{1}{\sigma^2}\left[(n + 4)\ln 2 + 2\ln(1/\epsilon)\right]$. By inserting Eq. (S74) into Eq. (S70), we can get

$$\left| \mathrm{Tr}\left(H\left(\bar{\rho} - \frac{I}{2^n}\right)\right) \right| \leq 2\sqrt{1 - \frac{16}{16 + \epsilon^2}} = \frac{2\epsilon}{\sqrt{16 + \epsilon^2}} \leq \frac{\epsilon}{2}. \tag{S75}$$

Bringing Eqs. (S69) and (S75) into Eq. (S67), we complete the proof of Corollary 2.1. $\qquad\square$

## D   Proof of Proposition 4

**Proposition 4.** *Consider a $K$-classification task with the data set $\mathcal{D}$ defined in Def. 3. If the encoding depth $D \geq \frac{1}{\sigma^2}\left[(n + 4)\ln 2 + 2\ln(1/\epsilon)\right]$ for some $\epsilon \in (0, 1)$, then the partial gradient of the loss function defined in Eq. (8) with respect to each parameter $\theta_i$ of the employed QNN is bounded as*

$$\left|\frac{\partial L(\boldsymbol{\theta}; \mathcal{D})}{\partial \theta_i}\right| \leq K\epsilon \tag{S76}$$

*with a probability of at least $1 - 2\mathrm{e}^{-M\epsilon^2/8}$.*

*Proof.* From the chain rule, we know

$$\frac{\partial L(\boldsymbol{\theta}; \mathcal{D})}{\partial \theta_i} = \frac{1}{KM}\sum_{m=1}^{KM}\frac{\partial L^{(m)}}{\partial \theta_i} = \frac{1}{KM}\sum_{m=1}^{KM}\sum_{l=1}^{K}\frac{\partial L^{(m)}}{\partial h_l}\frac{\partial h_l}{\partial \theta_i}. \tag{S77}$$

We first calculate $\frac{\partial L^{(m)}}{\partial h_l}$ as follows

$$\frac{\partial L^{(m)}}{\partial h_l} = \frac{\partial \sum_{k=1}^{K} y_k^{(m)}\left(\ln\sum_{j=1}^{K}\mathrm{e}^{h_j} - h_k\right)}{\partial h_l} = \begin{cases} \sum_{k=1}^{K} y_k^{(m)}\frac{\mathrm{e}^{h_l}}{\sum_{j=1}^{K}\mathrm{e}^{h_j}}, & l \neq k \\ \sum_{k=1}^{K} y_k^{(m)}\left(\frac{\mathrm{e}^{h_l}}{\sum_{j=1}^{K}\mathrm{e}^{h_j}} - 1\right), & l = k \end{cases} \tag{S78}$$

Since $\mathrm{e}^{h_l}/\sum_{j=1}^{K}\mathrm{e}^{h_j} \in (0, 1)$ and $\boldsymbol{y}^{(m)}$ is one-hot, we can get its upper bound as $\left|\frac{\partial L^{(m)}}{\partial h_l}\right| \leq 1$. Next, from the parameter-shift rule [45], we calculate $\frac{\partial h_l}{\partial \theta_i}$ as follows

$$\frac{\partial h_l}{\partial \theta_i} = \frac{1}{2}\left[\mathrm{Tr}\left(H_l U(\boldsymbol{\theta}_{+\frac{\pi}{2}})\rho(\boldsymbol{x}^{(m)})U^\dagger(\boldsymbol{\theta}_{+\frac{\pi}{2}})\right) - \mathrm{Tr}\left(H_l U(\boldsymbol{\theta}_{-\frac{\pi}{2}})\rho(\boldsymbol{x}^{(m)})U^\dagger(\boldsymbol{\theta}_{-\frac{\pi}{2}})\right)\right], \tag{S79}$$

where $\boldsymbol{\theta}_{+\frac{\pi}{2}}$ means adding $\frac{\pi}{2}$ to $\theta_i$ and keeping the others unchanged, and $\boldsymbol{\theta}_{-\frac{\pi}{2}}$ is similarly defined. If we define

$$\tilde{H}_l \equiv \frac{1}{2}\left[U^\dagger(\boldsymbol{\theta}_{+\frac{\pi}{2}})H_l U(\boldsymbol{\theta}_{+\frac{\pi}{2}}) - U^\dagger(\boldsymbol{\theta}_{-\frac{\pi}{2}})H_l U(\boldsymbol{\theta}_{-\frac{\pi}{2}})\right], \tag{S80}$$

then together with Eqs. (S77)-(S79), we could bound the gradient as

$$\left| \frac{\partial L\left(\boldsymbol{\theta};\mathcal{D}\right)}{\partial \theta_i} \right| \le \left| \frac{1}{KM} \sum_{m=1}^{KM} \sum_{l=1}^{K} \frac{\partial h_l}{\partial \theta_i} \right| = \left| \frac{1}{KM} \sum_{m=1}^{KM} \sum_{l=1}^{K} \mathrm{Tr}\left( \tilde{H}_l \rho(\boldsymbol{x}^{(m)}) \right) \right| \tag{S81}$$

$$\le \sum_{l=1}^{K} \left| \frac{1}{KM} \sum_{m=1}^{KM} \mathrm{Tr}\left( \tilde{H}_l \rho(\boldsymbol{x}^{(m)}) \right) \right| \tag{S82}$$

$$= \sum_{l=1}^{K} \left| \frac{1}{KM} \sum_{m=1}^{KM} \sum_{k=1}^{K} y_k^{(m)} \mathrm{Tr}\left( \tilde{H}_l \rho(\boldsymbol{x}^{(m)}) \right) \right| \tag{S83}$$

$$\le \sum_{l=1}^{K} \frac{1}{K} \sum_{k=1}^{K} \left| \frac{1}{M} \sum_{m=1}^{KM} y_k^{(m)} \mathrm{Tr}\left( \tilde{H}_l \rho(\boldsymbol{x}^{(m)}) \right) \right| \tag{S84}$$

$$\le \sum_{l=1}^{K} \frac{1}{K} \sum_{k=1}^{K} \epsilon. \tag{S85}$$

Here, it could be easily verified that the eigenvalues of $\tilde{H}_l$ (defined in Eq. (S80)) range in $[-1, 1]$ and $\mathrm{Tr}(\tilde{H}_l) = 0$. Then from Corollary 2.1, we could bound Eq. (S84) as Eq. (S85), i.e., for any $\epsilon \in (0, 1)$, provided that the encoding depth $D \ge \frac{1}{\sigma^2}\left[(n+4)\ln 2 + 2\ln(1/\epsilon)\right]$, we have $\left|\frac{\partial L(\boldsymbol{\theta};\mathcal{D})}{\partial \theta_i}\right| \le K\epsilon$ with a probability of at least $1 - 2\mathrm{e}^{-M\epsilon^2/8}$. This completes the proof of Proposition 4. $\qquad\square$