# OpenReview forum: "Concentration of Data Encoding in Parameterized Quantum Circuits"
_NeurIPS.cc/2022/Conference — NeurIPS 2022 Accept_

### Official Review · Reviewer_QanR · 2022-07-08

**Rating:** 7
**Confidence:** 3
**Soundness:** 3 good
**Presentation:** 3 good
**Contribution:** 2 fair

**Summary:**

This work analyzes the effect of classical data encoding for quantum machine learning models. Focusing on non-reuploading techniques, upper bounds are proved on the divergence between the encoded state and the maximally mixed state. These bounds show dramatic decreases in divergences with increases in number of qubits and depth. These bounds are then verified with a collection of numerical simulations.

=========================================

Note: after author response, rating moved from 6 to 7

**Questions:**

- A variety of gradient based optimizers have been used for QVCs, is there a reason why Adam was chosen over other methods?
- Based on the limitations presented in this work, is there any situation in which this style of angle embedding/encoding would be recommended?


**Limitations:**

The authors adequately addressed the potential negative societal impacts of their work.

**Strengths And Weaknesses:**

Pros:
- Understanding data encodings in quantum machine learning for classical data is critical to fully unlocking the potential of QML and this paper makes an important step in that direction
- The upper bounds proven show important limitations of certain encoding schemes and are practically relevant for QML with classical data
- The visual aids (figures and circuit diagrams) are helpful and make the information clearer
- The paper is generally well written and effective at conveying the point

Cons:
- Including some discussion of other encoding strategies would be beneficial. Understandably the focus is on one, but including other strategies in the empirical results would help practitioners make more informed decisions about encoding techniques.
- Defining cross entropy loss (eq. 8) seems unnecessary
- Depth 10 is very hard to see in Figure 6(c)
- Experiment code is not released, the authors acknowledge that it should be easy to reproduce, but releasing code would only help to ensure this

---

> ### Author Response · Authors · 2022-08-01
> **We thank the reviewer for their time and the appreciation of our paper as important, practically-relevant, and well-written. We address specific questions raised by the reviewer below.**
>
> > $\textbf{Q1:}$ ``A variety of gradient-based optimizers have been used for QVCs, is there a reason why Adam was chosen over other methods?''
>
> $\textbf{A1:}$ We choose Adam mainly because i) in the classic case (deep learning), Adam has excellent performance; ii) many relevant works use Adam to optimize the parameters in PQCs; iii) we indeed found Adam is stable and makes models converge to better accuracy than some other optimizers, like SGD.
>
> > $\textbf{Q2:}$ ``Based on the limitations presented in this work, is there any situation in which this style of angle embedding/encoding would be recommended?''
>
> $\textbf{A2:}$ Based on the theoretical analysis as well as the numerical results in this work, we conclude that this style of angle encoding with only shallow depth could be suitable and effective for QML, while higher depth will destroy more features in the original data. Hence, we think this encoding might be useful when the original data have been reduced to a suitable dimension by some pre-processing methods. We think a further study on designing new and hybrid methods of quantum data encoding based on the implications of our results will be very interesting and necessary.
>
> > $\textbf{Q3:}$ ``Including some discussion of other encoding strategies would be beneficial. Understandably the focus is on one, but including other strategies in the empirical results would help practitioners make more informed decisions about encoding techniques.''
>
> $\textbf{A3:}$ In the submitted manuscript, we focus on PQC-based encoding mainly because it is the mainstream encoding strategy in the NISQ era and employed by various research works (see reviews [1], [2] and [3]).
> We believe that pointing out the concentration issue of this encoding strategy is timely and significant in the current stage.
> We will also add some discussion of other popular encoding strategies, like amplitude encoding [4] or IQP encoding [5], in the final version if accepted.
>
> [1] Benedetti, Marcello, et al. "Parameterized quantum circuits as machine learning models." Quantum Science and Technology 4.4 (2019): 043001.
>
> [2] Bharti, Kishor, et al. "Noisy intermediate-scale quantum (NISQ) algorithms." arXiv preprint arXiv:2101.08448 (2021).
>
> [3] Cerezo, Marco, et al. "Variational quantum algorithms." Nature Reviews Physics 3.9 (2021): 625-644.
>
> [4] Schuld, Maria. "Supervised quantum machine learning models are kernel methods." arXiv preprint arXiv:2101.11020 (2021).
>
> [5] Havlíček, Vojtěch, et al. "Supervised learning with quantum-enhanced feature spaces." Nature 567.7747 (2019): 209-212.
>
> > $\textbf{Q4:}$ ``Defining cross entropy loss (eq. 8) seems unnecessary''
>
> $\textbf{A4:}$ Cross entropy loss is one of the most commonly used losses for classification tasks in machine learning. We define this loss aiming to describe a whole story. In particular, by bounding the gradient of the loss, we show how the concentration issue could severely limit the trainability of a QNN (see Proposition 4). One can also change it to other losses and obtain similar results.
>
> > $\textbf{Q5:}$ ``Depth 10 is very hard to see in Figure 6(c)''
>
> $\textbf{A5:}$ Thanks for this suggestion. We will bold the line of Depth 10 in the final version.
>
>
> > $\textbf{Q6:}$ ``Experiment code is not released, the authors acknowledge that it should be easy to reproduce, but releasing code would only help to ensure this''
>
> $\textbf{A6:}$ We thank the reviewer for this suggestion. We will release our code if accepted.

---

> > ### Comment · Reviewer_QanR · 2022-08-07
> > **Response to Author Response**
> >
> > The authors response to the comments and questions is appreciated. The clarifications have been helpful and the changes will improve the paper. I especially appreciate the response to Q3. I have updated the rating to reflect the positive impact of the changes.

---

### Official Review · Reviewer_GkYH · 2022-07-11

**Rating:** 7
**Confidence:** 3
**Soundness:** 3 good
**Presentation:** 4 excellent
**Contribution:** 3 good

**Summary:**

This paper raise an important issue for data encoding in PQC: concentration. The claim is theoretically and emprically verified. The paper is generaly well-written and the issue is indeed central in quantum machine learning.

**Questions:**

- How do you think about  learning a non-linear data encoding strategy if possiable?
- why specify ${\color{red} 3}nD$ for $\mathbf{x}$ in shape ?


**Limitations:**

not found

**Strengths And Weaknesses:**

Strengths:
- it is a important issue
- both theoretical and empirical verification
- well-written

Weaknesses:
- no solution provided for the mentioned issue.
- it does not consider various data encoding strategies.
- relatively weak experiment (but it is fine to evidence the concentration issue)

---

> ### Author Response · Authors · 2022-08-01
> **We would like to thank the reviewer for their time and their positive comments about the importance of our work. Below, we provide a brief response to the questions raised by the reviewer.**
>
> > $\textbf{Q1:}$ ``How do you think about learning a non-linear data encoding strategy if possible?''
>
> $\textbf{A1:}$ We would like to thank the reviewer for guiding us to think about the possibility of designing a non-linear encoding strategy. Due to the concentration issue of PQC-based encoding, we should design other suitable encoding strategies to avoid this issue. We think learning a non-linear data encoding strategy would be a potential candidate for a feasible solution. The reward-punishment mechanism in the learning process could be designed according to the quantum divergence we mentioned in this work. We will add the corresponding discussion in the revision.
>
>
> > $\textbf{Q2:}$ ``why specify 3nD for x in shape?''
>
> $\textbf{A2:}$ We thank the reviewer for raising this point. We specify $\textbf{x}$ as a $3nD$-dimensional vector because this matches our encoding strategy mentioned in Fig.~3, which has $n$ qubits and $D$ layers of $U3$ gates, where each $U3$ gate has $3$ parameters. We will make this point clearer in the final version.
>
> > $\textbf{Q3:}$ ``no solution provided for the mentioned issue.''
>
> $\textbf{A3:}$ The focus of our work is to point out a critical limitation of a family of mainstream encoding strategies in quantum machine learning, that is, the concentration of encoded data. We believe that our results are urgent and timely in the NISQ era, where encoding classical data into quantum ones is an essential step in quantum machine learning. Though we didn't dedicate a section to the solution to this concentration issue as it would be out of the scope of this work, our results can serve as guidance to the future design of PQC-based data encoding strategies. As we mentioned in the Discussion section, an apparent way is to keep the depth shallow by increasing the width of the PQC.
>
> > $\textbf{Q4:}$ ``it does not consider various data encoding strategies.''
>
> $\textbf{A4:}$ This is mainly because the PQC-based data encoding or angle encoding is the mainstream family of encoding strategies in the current stage (see [1], [2] and [3]). Hence, this work focuses on these strategies aiming to demonstrate their concentration issue. We think that pointing out the concentration issue of this encoding strategy is significant in the current stage. In the revision, we will also add some discussion of other popular encoding strategies, like amplitude encoding [4] and IQP encoding [5].
>
> [1] Benedetti, Marcello, et al. "Parameterized quantum circuits as machine learning models." Quantum Science and Technology 4.4 (2019): 043001.
>
> [2] Bharti, Kishor, et al. "Noisy intermediate-scale quantum (NISQ) algorithms." arXiv preprint arXiv:2101.08448 (2021).
>
> [3] Cerezo, Marco, et al. "Variational quantum algorithms." Nature Reviews Physics 3.9 (2021): 625-644.
>
> [4] Schuld, Maria. "Supervised quantum machine learning models are kernel methods." arXiv preprint arXiv:2101.11020 (2021).
>
> [5] Havlíček, Vojtěch, et al. "Supervised learning with quantum-enhanced feature spaces." Nature 567.7747 (2019): 209-212.

---

> > ### Comment · Reviewer_GkYH · 2022-08-09
> > **Thanks.**
> >
> > I have read the authors' replies. I am satisfied.

---

### Official Review · Reviewer_pq1y · 2022-07-11

**Rating:** 7
**Confidence:** 2
**Soundness:** 3 good
**Presentation:** 3 good
**Contribution:** 3 good

**Summary:**

This paper discussed some findings on the impact of data encoding in quantum NN

**Questions:**

* Figure 5 shows that the strategy in figure 4 would decay faster then the on in figure 2 as the qubit increases, is there any comparison on how would this affect training accuracy? Will strategy 4 also summer quick loss than strategy 2?

**Limitations:**

The authors have discussed the limitations of the proposed encoding method

**Strengths And Weaknesses:**

**Originality**:

* To my knowledge and understanding, this paper's findings seems original

**Quality**:

* The paper technically sounds correct and claims well supported by theoretical analysis and experimental results.
* Related works and background knowledge are covered and discussed.
* Experiments are conducted extensively in comparing different aspects for different encoding strategies, experimental results are thoroughly discussed.


**Clarity**:

* Structure wise, this paper is well written and organised.
* It would be easier to navigate if encoding strategies can have names instead of the one in F2 or F4.



**Significance**:
The finding suggests that different encoding strategies have impact on training accuracy as it affects feature preservation.

---

> ### Author Response · Authors · 2022-08-01
> **Many thanks for the reviewer's positive assessments on the importance and originality of our work. We appreciate the valuable comments and would like to response to the questions raised by the reviewer.**
>
> > $\textbf{Q1:}$ ``Figure 5 shows that the strategy in figure 4 would decay faster then the on in figure 2 as the qubit increases, is there any comparison on how would this affect training accuracy? Will strategy 4 also summer quick loss than strategy 2?''
>
> $\textbf{A1:}$ Yes. As the main purpose of our work is to point out the concentration issue, we only put the results for the strategy in Fig. 4 to verify this. But according to our numerical findings, the values of the actual quantum divergences directly affect the training and testing accuracy, which means strategy 4 will result in higher loss more quickly than strategy 2 as encoding depth grows.
>
> > $\textbf{Q2:}$ ``It would be easier to navigate if encoding strategies can have names instead of the one in F2 or F4.''
>
> $\textbf{A2:}$ Thanks for this suggestion. We will name these strategies in the final version if accepted.

---

> > ### Comment · Reviewer_pq1y · 2022-08-09
> > **Response to Rebuttal**
> >
> > I have read the authors' response and would like to thank them for providing the update. I am satisfied with it.

---

### Meta-Review · Area_Chair_Thny · 2022-08-24

**Recommendation:** Accept
**Confidence:** Certain

**Metareview:**

This submission studies the effect of data encoding from quantum machine learning models. All reviewers agreed on the significance of this work and recommended acceptance.

**Award:**

No

---

### Decision · Program_Chairs · 2022-09-14

Accept